# Mottness versus unit-cell doubling as the driver of the insulating state in 1$T$-TaS$_2$

C. J. Butler [1✉], M. Yoshida [1], T. Hanaguri [1✉] & Y. Iwasa[1,2]

If a material with an odd number of electrons per unit-cell is insulating, Mott localisation may be invoked as an explanation. This is widely accepted for the layered compound 1$T$-TaS$_2$, which has a low-temperature insulating phase comprising charge order clusters with 13 unpaired orbitals each. But if the stacking of layers doubles the unit-cell to include an even number of orbitals, the nature of the insulating state is ambiguous. Here, scanning tunnelling microscopy reveals two distinct terminations of the charge order in 1$T$-TaS$_2$, the sign of such a double-layer stacking pattern. However, spectroscopy at both terminations allows us to disentangle unit-cell doubling effects and determine that Mott localisation alone can drive gap formation. We also observe the collapse of Mottness at an extrinsically re-stacked termination, demonstrating that the microscopic mechanism of insulator-metal transitions lies in degrees of freedom of inter-layer stacking.

[1] RIKEN Center for Emergent Matter Science, 2-1 Hirosawa, Wako, Saitama 351-0198, Japan. [2] Quantum-Phase Electronics Center and Department of Applied Physics, The University of Tokyo, 7-3-1 Hongo, Bunkyo-ku, Tokyo 113-8656, Japan. ✉email: christopher.butler@riken.jp; hanaguri@riken.jp

The origin of the spectral gap in many insulating materials is difficult to determine because as well as the simple band theoretic criterion of a completely filled valence band, electron–phonon interactions, strong electronic correlations[1,2] and other mechanisms generally can coexist and may all play some role. This is true in the decades-old charge density wave (CDW) compound $1T$-TaS$_2$, for which the debate over the nature of the low-temperature insulating state has only intensified in recent years[3–7]. Although the proximate cause of this insulating state is under debate, its precursor is known to be an electron–phonon driven commensurate CDW (C-CDW) phase. The undistorted high-temperature atomic structure of $1T$-TaS$_2$ is shown in Fig. 1a. Below ~350 K the Ta lattice within each layer undergoes a periodic in-plane distortion in which clusters of 13 Ta ions contract towards the central ion of the cluster, forming a Star-of-David (SD) motif[8]. Upon cooling below ~180 K this pattern locks in to become commensurate with the atomic lattice and long range order emerges, described as a triangular $\sqrt{13} \times \sqrt{13}\, R13.9°$ C-CDW pattern, depicted in Fig. 1b. Within each of the SD clusters 12 of the Ta $5d$ orbitals at the periphery form six filled bands and leave a CDW gap[9], stabilising the distortion. The remaining orbital, according to band theory, should form a half-filled band, and the experimentally observed insulating behaviour is usually attributed to its localisation at the SD centre by strong electron-electron (e–e) interactions[10,11]. From this foundation it has been suggested that, since a Mott state in $1T$-TaS$_2$ realises a triangular lattice of localised $S = ½$ spins, it

might host a quantum spin liquid (QSL), an unusual phase of quantum electronic matter in which, due to geometric frustration and quantum fluctuations, the spins refuse to magnetically order even down to $T = 0$ K[12–16].

The Mott state thought to exist in $1T$-TaS$_2$ is different from ordinary Mott insulators such as NiO in that electrons localise not at the sites of the atomic crystal, but at the sites of the electronic crystal, the lattice of SD clusters, and so it is called a cluster Mott insulator. As the SD clusters must be centred on Ta sites the three-dimensional (3D) structure formed from the layering of 2D charge order lattices can be described with stacking vectors $\mathbf{T}$ composed of the underlying Ta lattice vectors. There are five symmetrically inequivalent stacking vectors, which may be collected into only three groups according to their length: $\mathbf{T}_A = \mathbf{c}$, $\mathbf{T}_B = \pm\mathbf{a} + \mathbf{c}$, and $\mathbf{T}_C = \pm 2\mathbf{a} + \mathbf{c}$ (or equivalently, $\mp 2\mathbf{a} \mp \mathbf{b} + \mathbf{c}$)[7].

The impact of this stacking degree of freedom on the electronic structure of $1T$-TaS$_2$ was largely neglected until Ritschel et al.[17] predicted, using ab initio calculations, that different inter-layer stacking patterns could result in a metallic phase (for $\mathbf{T}_C$ stacking) as an alternative to the well-known insulating phase (previously assumed to have $\mathbf{T}_A$ stacking). Going further, Ritschel et al.[5] and Lee et al.[7] recently challenged the rationale by which $1T$-TaS$_2$ was thought to be a Mott insulator, showing that if the stacking alternates between vectors $\mathbf{T}_A$ and $\mathbf{T}_C$ as previously suggested[18–20], such that the new supercell includes two SD clusters, ab initio calculations predict an insulator without the need to invoke strong e–e interactions. (It has been established

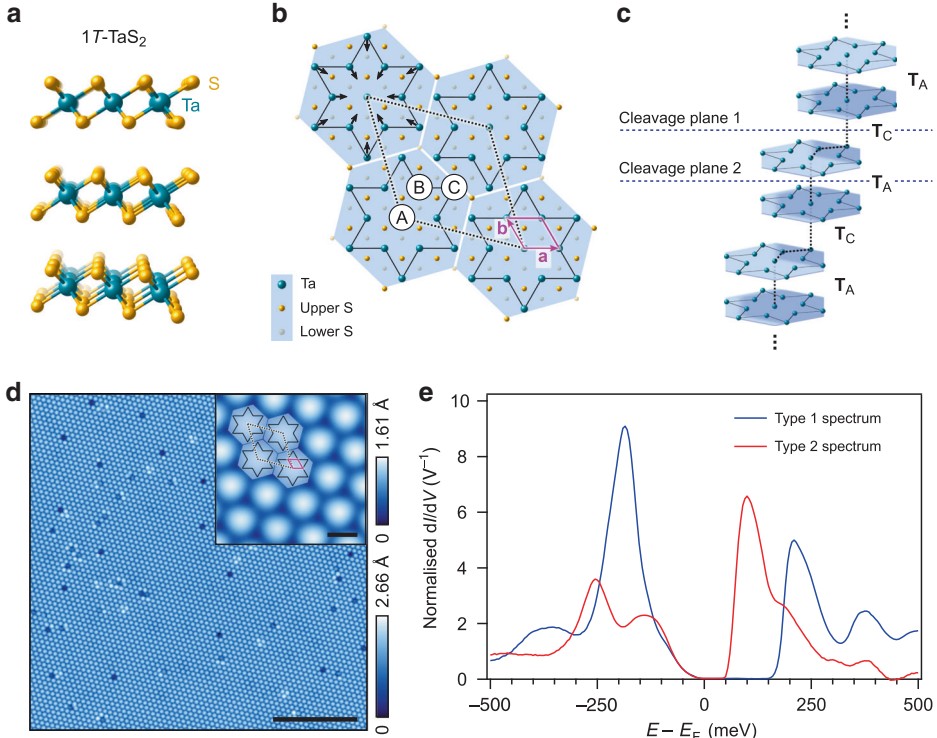

**Fig. 1 Overview of charge order, inter-layer stacking and cleaved surfaces in $1T$-TaS$_2$. a** The quasi-2D undistorted structure of $1T$-TaS$_2$. **b** The supercell describing the periodic SD distortion within a single $1T$-TaS$_2$ layer. The solid purple and dashed black rhombuses mark the 2D projections of the undistorted atomic unit-cell, and the supercell after onset of the C-CDW, respectively. The labels A, B, and C denote the possible sites atop which successive SD clusters can stack. **c** The SD stacking pattern currently discussed (S not shown), with two SDs per cell and two distinct cleavage planes, 1 & 2. **d** Typical STM topography taken at a vacuum-cleaved $1T$-TaS$_2$ surface ($V = 250$ mV, $I_{set} = 500$ pA, scale bar 20 nm). The inset shows the correspondence between the topographic modulation and the SD cluster lattice (scale bar 1 nm). **e** Examples of conductance spectra of the two types observed at multiple cleaved surfaces. Typically, one type of spectrum or the other appears uniformly (except in the vicinity of defects) over ~1 μm areas, unless a step-terrace morphology is observed. The prominent conductance peaks at around 200 meV and −200 meV in the previously reported Type 1 spectrum have usually been identified with the upper and lower Hubbard bands. It will be shown below that the Type 1 & 2 spectra correspond to surfaces formed by cleavage at planes 1 & 2, respectively.

that the bulk stacking structure likely alternates between $T_A$ and a vector drawn randomly from three versions of $T_C$ related by rotations of 120°, in a partially disordered pattern – see Supplementary Note 1. The dimerisation of the stacking structure into bilayers, and the disorder, have also been discussed in the interpretation of recent experimental works[21,22]). Put simply, if the electronic unit-cell contains two SDs, the total number of electrons per cell is even, leaving the highest occupied band filled and allowing an insulator without invoking Mott. This introduces serious complication into the understanding of the insulating state in $1T$-$TaS_2$, and potentially undermines the foundations on which recent suggestions of a QSL state are built[13].

Here, we report on low-temperature scanning tunnelling microscopy (STM) measurements, which appear to confirm the premise described above: a unit-cell doubling inter-layer stacking pattern is indeed realised in $1T$-$TaS_2$. Despite this, we see that a spectral gap persists at a surface where dimer-like inter-layer pairing is broken, which is unexpected unless e–e interactions play a significant role. We also show that for such an unpaired layer of SD clusters, a small change in stacking with respect to the underlying layer yields a metallic surface, suggesting that inter-layer effects underpin the microscopic mechanism of the material's metal-insulator transitions[23–29].

## Results

**Observation of spectroscopically distinct surfaces.** A consequence of the $T_A$, $T_C$, $T_A$, $T_C$...(henceforth ACAC) stacking pattern is that there are two cleavage planes, as indicated in Fig. 1c, yielding two inequivalent surfaces amenable to investigation using STM. One plane is located between one $T_A$-stacked bilayer (BL) and another, and the other plane splits a single BL, leaving unpaired ($T_C$-stacked) layers. In this work, samples were cleaved, transferred to the STM and measured at temperatures far below the transition temperature at which the C-CDW sets in (i.e., far below ~180 K, see "Methods"), and the bulk structure of the CDW should be preserved such that measurements on a large number of cleaved surfaces may show evidence of the ACAC pattern. Eight platelets of $1T$-$TaS_2$ were cleaved multiple times each, for a total of twenty-four investigated surfaces, the topographic image for one of which appears in Fig. 1d. Conductance spectra were acquired at defect-free locations on each sample. Spectra showing a gap in the density of states (DOS) of ~150 meV, broadly consistent with those shown in previous STM reports[9,28–30] were observed on 18 of the 24 surfaces (similar to the blue curve labelled Type 1 in Fig. 1e). The prominent conductance peaks at around 200 meV and −200 meV have usually been identified with the upper and lower Hubbard bands (UHB and LHB), respectively, characteristic of the Mott insulating state[31]. A different form of the DOS, with a smaller gap of 50–60 meV, was observed on the remaining six (Type 2, the red curve in Fig. 1e; we only consider the spectra acquired in the regions where the STM tip first arrived at the sample surface. Additional data and discussion elucidating the distinct spectral features of each surface, and their spatial distributions, are provided in Supplementary Note 2). We tentatively attribute the appearance of these two forms of DOS to the surfaces created by the two cleavage planes of the bulk stacking pattern. However, more information is needed to definitively assign each form of DOS to each cleaved surface, and we return to this below. If the number of cleavage planes of each type throughout the sample is roughly equal, as we assume, the deviation of the observed ratio from 1:1 may correspond to a difference in their associated surface formation energies, and the resulting rarity of the small-gap surface may explain its absence in previous reports. A small number of instances were observed where the two types of surface appeared side-by-side, for example on either side of a domain wall[30] in the CDW pattern (see Supplementary Note 3). The tip-height dependence of each of the spectra was investigated, showing that there is no height dependent crossover between one type of spectrum and the other (see Supplementary Note 4).

**Determination of inter-layer stacking.** Most revealingly, the two types were also observed side-by-side where single-layer steps allowed the simultaneous observation of multiple $TaS_2$ layers. Figure 2a shows a topographic image of three terraces, with the upper terrace featuring a domain wall (marked with a dark blue dotted line), so that four distinct regions are observed (labelled Regions 1–4). Tunnelling spectroscopy acquired along a path spanning the long axis of the topographic image (marked with a red-tinted rectangle) shows changes in the DOS spectrum upon each transition between regions (Fig. 2b). Representative spectra taken at a point within each of the four regions are shown in Fig. 2c–f. Region 2 shows a finite DOS at $E_F$, reminiscent of the so-called metallic mosaic phase, which has been created locally using STM-induced voltage pulses[28], with inter-layer stacking effects suggested as a possible explanation[29]. Briefly postponing the discussion of this metallic phase, we first note that the form of the DOS in the other three regions is seen to alternate from one terrace to the next, from a small gap (Region 1) to a large gap in the middle terrace (Region 3), and to the small gap again at the lowest terrace (Region 4). This alternating sequence is consistent with that expected for the ACAC stacking shown in Fig. 1c (another, similar instance of the switching of electronic structure from one type to the other across a single-layer step is shown in Supplementary Fig. 4.)

With a view to establishing which type of surface corresponds to which of the cleavage planes in the ACAC stacking pattern, we note that single-layer steps should result in an in-plane displacement, or phase jump $\Delta\phi$, of the 2D projected CDW pattern from one terrace to the next, which should alternate between zero and non-zero (specifically ±2$a$, or equivalently, $\mp 2a \mp b$, the in-plane projection of $T_C$), as is shown in Fig. 3a. The step between Regions 3 & 4 realises the former case: in Fig. 3b, the SD centres in these two regions are highlighted with an array of white dots, showing clearly the absence of an in-plane displacement and hence indicating that in Region 3 the stacking pattern terminates with an intact BL at the surface, and that in Region 4, an unpaired layer of SD clusters remains. The array of white dots is extended over to the left-hand-side of the image. We make the assumption that neither of the lower layers host a domain wall hidden below the uppermost layer, so that overlaying the centres of the SD lattices of the upper terrace with this extended reference lattice shows the approximate in-plane shift between layers. We infer from the change of electronic structure between Regions 1 & 2 that the observed domain wall exists only in the uppermost layer (the ordinary (intrinsic) domain walls, which penetrate through multiple layers usually are not accompanied by such a change in electronic structure. See Supplementary Note 3). Additional atomically resolved topographic imaging in Region 1 (see Supplementary Note 6) is used to determine the orientation of the SD clusters, depicted in the zoom-in images of Fig. 3c, d. The in-plane components of the stacking vectors for Regions 1 & 2 are then discernible. In Region 1 the stacking vector is shown to correspond closely to $T_C$, as expected for the ACAC stacking pattern, with a discrepancy well below one atomic lattice constant. The stacking vector between the metallic region, Region 2, and the underlying BL is also determined, and corresponds closely to the $T_B$. As this type of metallic region was observed nowhere else throughout measurements on 23 other samples, we posit that it is an outcome of an

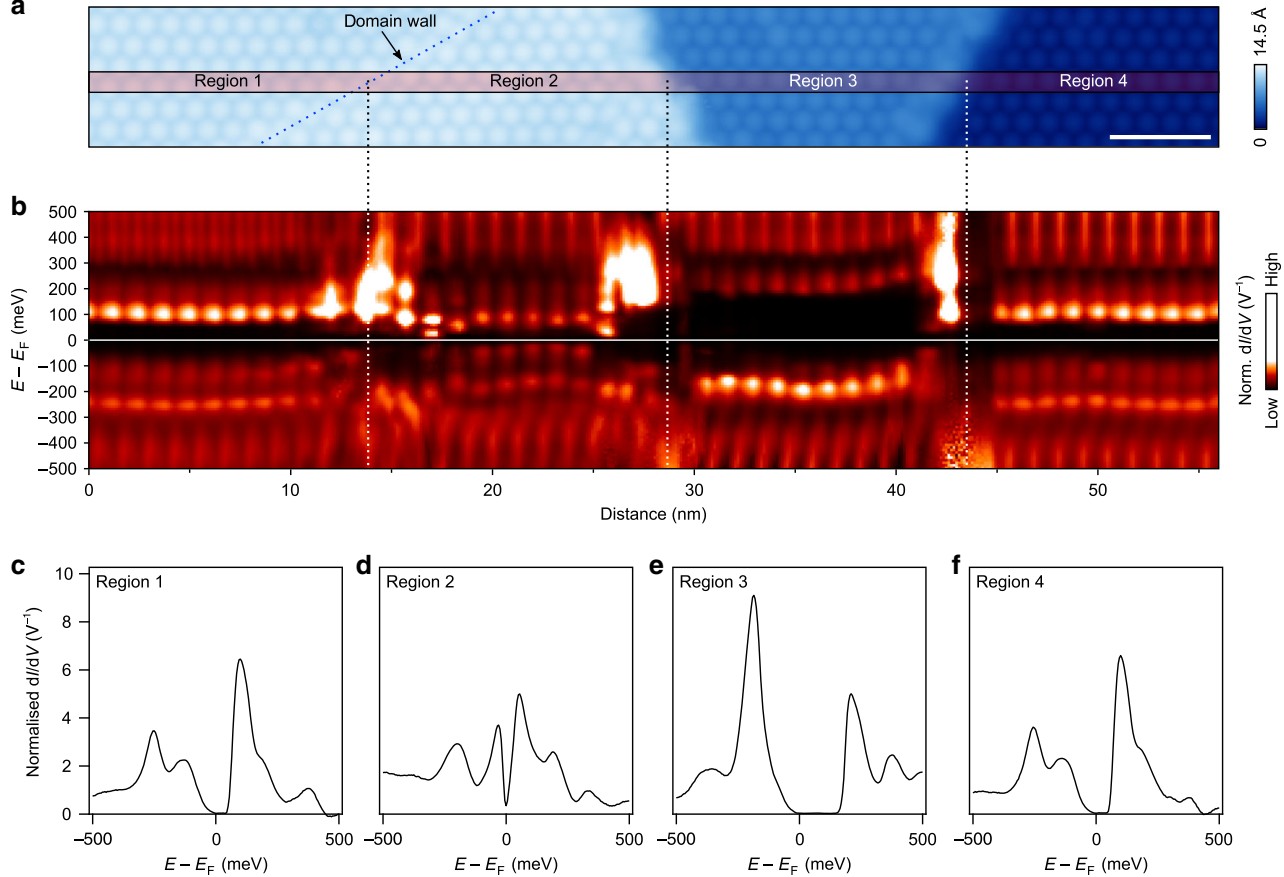

**Fig. 2 Conductance spectra across a step-terrace formation. a** A topographic image showing three terraces separated by two steps ($V = 250$ mV, $I_{set} =$ 125 pA, scale bar 5 nm). The uppermost terrace at the left-hand-side is further divided into two regions separated by a domain wall, Regions 1 & 2. **b** Spectroscopy along the path marked by the red-tinted rectangle in **a**, averaged over the rectangle's short axis (the width of the rectangle was chosen so as to average over, approximately, the vertical projection of one CDW period. The raw conductance curves from which this plot was obtained are shown in Supplementary Note 5). **c–f** Representative spectra collected in each of the four regions, 1–4, shown in **a**. Leaving aside the metallic Region 2, the type of spectrum alternates layer-by-layer.

extrinsic process during formation of the step-edge, such as a brief lifting of the uppermost TaS$_2$ layer and re-stacking of its CDW into a metastable configuration.

## Discussion
Taken together, the observations presented in Figs. 2 and 3 form a picture that is indeed consistent with the previously suggested ACAC stacking pattern, and allow us to establish a link between the surface CDW configuration and the surface local DOS, as summarised in Fig. 3e. The spectrum observed in Region 3 with the relatively large gap of ~150 meV has been reported in multiple STM works. We now suggest that this spectrum signifies a surface where the 3D CDW terminates with the T$_A$-stacked BL intact, and can be thought of as the bulk-like termination (i.e., without major structural or electronic reconfiguration upon formation of the surface). The fact that this surface was the most common outcome from cleavage (18 out of 24) indicates the energetic favourability of cleaving between BLs, rather than through them, suggesting non-negligible intra-BL bonding, or dimerisation.

The surface of unpaired SD clusters (Regions 1 & 4) represents a new and perhaps qualitatively distinct system, which may allow us to disentangle the role of strong e–e interactions from that of unit-cell doubling in the electronic structure. First, we note that if the BL-stacked bulk charge order can be considered as dimerised, then breaking the dimerisation by terminating the structure with

a layer of unpaired clusters should be expected to leave a metallic surface state, at least in the absence of e–e correlations. This remains true even if significant inter-layer hopping is present, as long as the intra-dimer hopping, in this case across the T$_A$-stacked interface, is greater than that across the T$_C$-stacked interface between the uppermost layer and those below (a comprehensive argument is given in Supplementary Note 7). Hence, the spectral gap observed in the unpaired SD layer, where an otherwise metallic surface state is expected, may be attributed to Mott localisation. It is possible that the anisotropic coordination environment of the T$_C$-stacked clusters results in a hierarchy of varying nearest-neighbour orbital overlaps, which could explain the detailed spectroscopic features observed. The resulting modification of bandwidth may also be the cause of the reduced size of the Mott gap.

The apparent role of Mottness in the unpaired SD layer also suggests that the metallic state also observed (Region 2 in Fig. 2) is properly described as a Mottness-collapsed state. The orbital localised at the SD centre is most likely a Ta $5d_{z^2}$[9,32], and due to its anisotropic projection perpendicular to the plane, it is reasonable to expect the inter-layer orbital overlap to be very sensitive to the lateral displacement between layers. In this case, upon changing the stacking from T$_C$ to T$_B$, we may speculate that the inter-layer overlap increases beyond the threshold for breakdown of the Mott state (further discussion of this effect is presented in Supplementary Note 8).

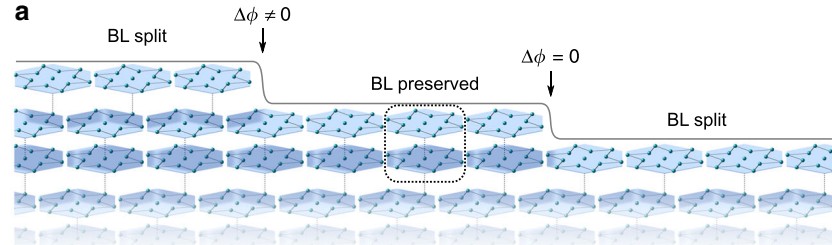

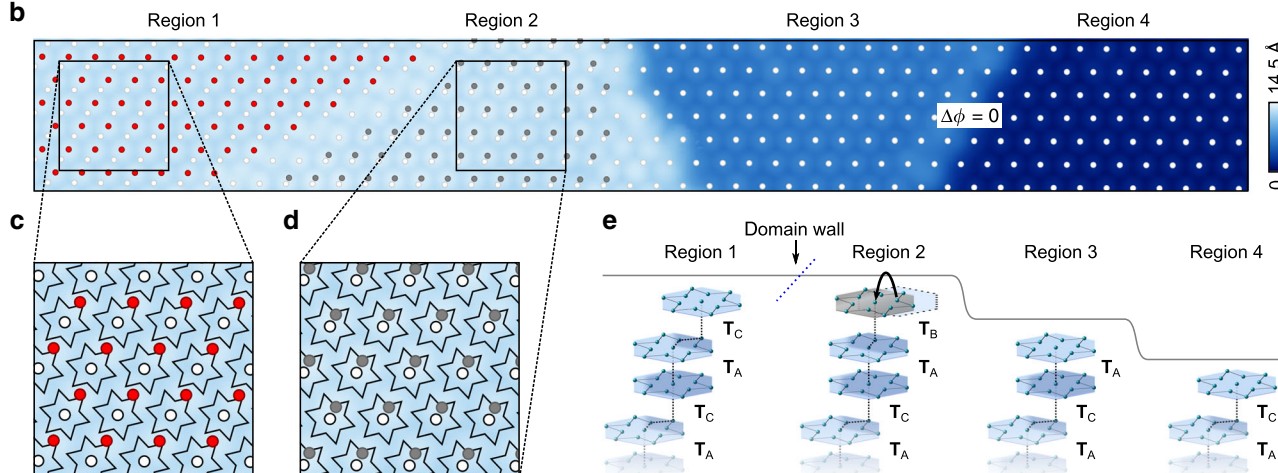

**Fig. 3 Identification of inter-layer stacking vectors. a** A schematic showing the in-plane displacements or phase shifts $\Delta\phi$, associated with steps of a pristine cleaved step-terrace morphology resulting from the ACAC structure. $\Delta\phi$ alternates between zero and non-zero. **b** The same STM topography shown in Fig. 2, with the SD cluster centres in Regions 3 & 4 marked with white dots. The lower two terraces are shown to be in phase with each other, indicating that they belong to the same BL. The lattice from the lower two terraces is extended to the left-hand-side of the field-of-view. The cluster centres in Regions 1 & 2 are shown with red and grey dots, respectively. **c, d** The relative positions of the top terrace clusters with respect to the SD motifs of the underlying BL. **e** Schematic depictions of the inter-layer stacking configuration for each region, to which we attribute the DOS spectra shown in Fig. 2. The layer in Region 2 that we interpret as extrinsically re-stacked is shown in grey.

Overall, these results emphasise the profound importance of inter-layer effects in the nature of the insulating state in $1T$-TaS$_2$, and also in its metal-insulator transitions. They also indicate that while the BL-stacked bulk structure of $1T$-TaS$_2$ may satisfy the criteria for a simple band insulator[5,7], this does not preclude the presence of strong e–e correlations, and these have been evidenced in the recent observation of doublon excitations characteristic of a Mott state[6]. Indeed, it has previously been shown that a system described by a two-layered Hubbard model with inter-layer hopping $t_\perp$ can exhibit a continuous crossover between the Mott and band insulating regimes[33]. Hence, for the paired surface and the BL-stacked bulk, with significant intra-BL overlap, the two regimes may not be meaningfully distinct. The effects of strong e–e correlations, while latent in the bulk, become prominent at surfaces where the layer dimerisation breaks down. We also note that an insulating state in $T_C$-stacked, unpaired layers also leaves some room for the persistence, at surfaces and possibly bulk stacking faults, of a QSL ground state despite possible inter-layer singlet formation within the bulk BLs[13]. Questions arise about the detailed mechanisms in play at the distinct surfaces observed, as well as between layers in the ACAC-stacked bulk—Why does the surface of unpaired clusters have a smaller gap than the paired layer, and what explains the details of its spectral shape? Exactly why does Mottness breakdown leading to metallicity for $T_B$ stacking? Further investigations of the detailed behaviour of the 3D electronic correlations in $1T$-TaS$_2$, especially at its surface terminations and also in few-layer or monolayer form, may prove fruitful in understanding the nature of its insulating state and its potentially useful metal-insulator transitions.

## Methods

**Synthesis and low-temperature cleavage of samples.** Crystals of $1T$-TaS$_2$ were synthesised using a chemical vapour transport method described previously[34], with 2% excess S. Samples were cleaved in ultra-high vacuum (~$10^{-10}$ Torr) at 77 K. Importantly, the head of the transfer rod that was subsequently used to insert samples into the pre-cooled STM was cooled along with each sample to a temperature near 77 K, and samples were inserted within a few tens of seconds after cleavage. We suggest that the maintenance of temperature far below the onset of the C-CDW phase (~180 K) is important for achieving cleaved surfaces that yield information about the pre-formed bulk stacking structure. For samples cooled through the C-CDW transition only after cleavage, a surface that retains information about the bulk stacking order may not be expected, since the presence of the surface may set boundary conditions on the CDW formation. In most previous STM reports showing d$I$/d$V$ data that can be compared against those presented here, the temperature of cleavage was either reported to be room temperature[29,30,35], or else was unspecified[28,36]. To our knowledge, only in one work was it specified that the cleavage temperature was below the transition temperature into the C-CDW phase[9], and there a spectrum was shown that appears very similar to the the Type 1 spectrum presented in the current work (Fig. 1e). Moreover, in the present work, where a bulk-like termination of the 3D charge order was achieved, the Type 1 spectrum was still seen for most cleaved surfaces. Thus, our results are not inconsistent with previous reports.

**STM measurements.** The STM used was a Unisoku 1300 low-temperature STM system, of which the STM head has been replaced with a homemade one[37]. All measurements were performed at a temperature of 1.5 K. STM tips were formed using electro-chemical etching of tungsten wire, and after insertion into UHV, were cleaned and characterised using field ion microscopy followed by careful conditioning on a clean Cu(111) surface. STM topography images were collected in constant-current mode. For conductance curves and spectroscopic mapping on $1T$-TaS$_2$ surfaces, the lock-in technique with a bias modulation of amplitude $V_{mod} = 10$ mV and frequency $f_{mod} = 617.3$ Hz was used.

The d$I$/d$V$ spectroscopy data were normalised according to the $I(V)$ value at $V = -500$ mV for Fig. 2b and at $V = +500$ mV elsewhere. This somewhat compensates for the large difference in raw signal intensity caused by differing tip-

sample distances while scanning at $V = 250$ mV over areas with large or small spectral gaps, and is performed only to aid visualisation. The raw d$I$/d$V$ curves for Fig. 2b are shown in Supplementary Fig. 7.

## Data availability
The data that support the findings presented here are available from the corresponding authors upon reasonable request.

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

## Acknowledgements
We are grateful to Y. Kohsaka, T. Machida, J. Lee, P. A. Lee, M. Hirayama and R. Arita for helpful discussions. C.J.B. gratefully acknowledges support from RIKEN's SPDR fellowship. This work was supported in part by JSPS KAKENHI grant numbers JP18K13511, JP19H00653, JP19H01855 and JP19H05602.

## Author contributions
T.H. and Y.I. conceived the project, and M.Y. and Y.I. synthesised the 1$T$-TaS$_2$ crystals. C.J.B. performed the STM measurements with assistance from T.H., and prepared the manuscript with input from all authors.

## Competing interests
The authors declare no competing interests.
