## [Peer Review File · Nature Communications]

Reviewers' comments:

Reviewer #1 (Remarks to the Author):

Butler et al. report the tunneling spectroscopy of the two different surface terminations in the commensurate ground state of 1T-TaS₂ in order to distinguish between the band and correlated origin of the insulating spectral gap. Prior structural studies found a dimerization in the interlayer direction leading to a simple band-insulator picture. The isolated bilayer was suggested to behave either as a band metal or a band insulator depending on the stacking according to the more rigorous first-principle studies. The authors skillfully test these ideas by mapping the spectra on a sequence of crystal steps and reveal instead that both terminations are gapped, and the gap size changes regularly with the period of two layers. The manuscript thus presents the compelling evidence towards the predominantly correlated origin of the insulated state.

The manuscript is timely, the results are novel and finally elucidate the role of dimerization in electronic spectra, removing the apparent inconsistency between the band and correlated pictures. At the same time, they raise further questions to be (re)considered about the role of the dimerization for spin-related properties and Mottness collapse in the metal-to-insulator switching. I am happy to recommend this nice paper for publication in Nature Communications given the authors properly address the issues listed below:

1. The biggest concern is related to the fact that the tunneling spectroscopy/microscopy is a surface technique with a finite penetration depth. This raises the two points (a-b) listed below:

1a) The authors can only guess the David star positions in the layers below the one they image. The ambiguity originates from the finite correlation length (~ 3 to 10 unit cells) of the dimerization and large number of stacking faults, as shown in the structural studies (Ritschel et al., Tanda et al.). Therefore, the extension of the CDW lattice from the region 4 to the regions 1&2 in Figure 3 is speculative and should be clearly marked as such. For example, the stacking fault in the layer of Reg. 4 can exist beneath the layer of Reg. 1 or 2 and the authors apparently cannot see it using STM. The ambiguity and its consequences should be clearly discussed in the text.

1b) The tunneling current can reach one layer beneath, as has been reported before and also stated by the authors when discussing the buried domain wall in Suppl. Fig. 1. Therefore, the spectra of the two neighboring layers would be convoluted, if no precautions are taken. I suggest the authors vary the tip-sample separation in the spectroscopy measurements to reduce the penetration depth and thus extract the clear spectra for the two terminations. I would like to see the set of spectra measured with the different tip-sample separations. I would expect the transition from clear to mixed shape (like Reg. III in Suppl. Fig. 1d) as the tip is moved closer.

2. The authors discuss Mottness collapse based on the U/t ratio. Surprisingly, the shortest inter-cluster distance is in the interlayer direction: the in-plane distance is $a\sqrt{13} \sim 1.2$ nm (a – the atomic period), the out-of-plane distance is $|a + b + c| \sim 0.83$ nm (for ACAC... stacking, c – the interlayer period). It is also known, that the Mottness collapse occurs inside the domain wall (e.g. Cho et al., Skolimowski et al., Fig. 2 and Suppl. Fig. 1 in the present manuscript), where SD clusters are moved closer e.g. by one atomic period inside the plane, resulting in the distance of $3a \sim 1\text{nm} > 0.83\text{nm} \sim |a + b + c|$. Why the Mott state does not collapse in the ACAC... stacking then? Careful calculations are necessary in this case, so I suggest the authors to be more careful in making implications from the simple U/t argument in such a complex multi-atom cluster with further complications like the interlayer S-S bonding etc. (cf. Lee et al.).

3. Are there any evidence for a strain or SD cluster distortion in/around the smaller gap areas (e.g. Reg. II in Suppl. Fig. 1a)?

Can the local stacking on the edge or near the domain wall be manipulated by strain (e.g. from the tip)?

I suggest briefly commenting these questions in the text if possible.

4. It is not clear from the present density plot of the dI/dV curves whether the gap is present very close to the step? I suggest showing the raw curves in the supplement.

Small/technical comments:

5. It would be better to use easily distinguishable colors for the atomic and cluster unit cells in Fig. 1b.

6. I suggest the $T_C = a + b + c$ notation to be dubbed with $T_C = 2a + c$ to be consistent with a number of the previous theoretical studies.

7. I would like to bring the authors' attention to the following three papers dealing with the three-dimensional stacking:

a) A. S. Nغانkeu et al. Quasi-one-dimensional metallic band dispersion in the commensurate charge density wave of 1T-TaS₂. Phys. Rev. B 96, 195147 (2017)

<https://doi.org/10.1103/PhysRevB.96.195147>

b) P. Darancet et al. Three-dimensional metallic and two-dimensional insulating behavior in octahedral tantalum dichalcogenides. Phys. Rev. B 90, 045134 (2014) <https://doi.org/10.1103/PhysRevB.90.045134>

c) D. Svetin et al. Three-dimensional resistivity and switching between correlated electronic states in 1T-TaS₂, Sci Rep 7, 46048 (2017) <https://doi.org/10.1038/srep46048>

Reviewer #2 (Remarks to the Author):

This paper discusses STM/STS measurements on a new cleaving plane of bulk 1T-TaS₂. Through STM imaging of a multiple-step structure on the TaS₂ surface, the authors clearly identify three types of CDW stacking orders whose STS spectra yield very different electronic structure. The important observation here is that when the cleavage occurs between an on-top-stacked bilayer (BL) the STS still has a gap at E_F . The authors view this split BL as an "unpaired" layer and claim that it should not have a gap without invoking Coulomb repulsion (i.e., Mott physics). The STM/STS data are of high quality and this is a very interesting observation; however, I find the interpretation not convincing enough to justify publication at this point, as described below.

The crux of the authors' argument is an unpaired surface layer hosting an odd number of electrons per unit cell which allows trivial gap opening mechanisms to be ruled out. However, this argument is not so convincing since the unpaired top layer has finite interlayer coupling to the layers underneath and can't be simply viewed as a decoupled layer. I believe it is important for the authors to show better evidence that for this stacking (and in the absence of correlation) there would indeed be a metallic surface state with an odd electron filling even though some interlayer coupling still exists. For a Mott-Hubbard gap to open this metallic surface state band has to have a small bandwidth and be half-filled, and the authors should provide more evidence that these conditions are fulfilled. Can the authors perform DFT simulations of this structure using a finite slab model to investigate the surface electronic properties? The problem here is that ARPES measurements and simulations have shown dispersion in the z -direction (PRB 96, 195147(2017), PRB 90, 045134 (2014)) and theoretical calculation of bulk "Tc-stacked" TaS₂ have also exhibited metallic behavior with a ~ 400 meV bandwidth (PRL 122, 106404 (2019)). Both of these results suggest that there can be significant coupling and significant bandwidth for Tc-stacked layers, which goes against the authors' interpretation of a decoupled top layer with very narrow bandwidth (i.e., narrow enough to drive a Mott transition). I feel that this paper shows some very beautiful data, but that this issue must be addressed a little more thoroughly in order to support the "Mottness" claim mentioned in the paper's title.

I have additional comments regarding this manuscript:

1. Since both bulk TaS₂ and the reported surface state are insulating, the measured STS gap size can be dramatically affected by band bending effects from the tip and other sample-related origins (as shown, for example, on Si(111) in PRB 80, 125326 (2009)). I believe this type of analysis is necessary to identify the true gap size of the unpaired layer. Also, how large is the lateral band bending effect in different domains in Fig. 2b?

2. Many people have measured bulk 1T-TaS₂ with STM, and it is a little surprising (and also quite interesting) to see a previously unreported cleavage plane. Did the authors do anything special to expose these cleaving planes? Is there anything special about their crystals and how is their crystal quality compared to other published results?

Reply Letter

We sincerely thank both Referees for their insightful and helpful comments. Below we describe our revisions to the manuscript in response to these comments, including additional experimental results which were suggested. Throughout this Reply Letter we reproduce some text from the Referees' remarks and highlight these in red, and where we make additions or revisions to the text of the manuscript, those are highlighted in blue.

This Reply Letter is followed by a comprehensive List of Changes to the manuscript, where the revisions and additions are listed (roughly) in order of their appearance in the manuscript.

Reply to Referee 1

First we thank the Referee for recognizing the novelty and significance of the current experimental results. The Referee's main concern, expressed in two parts, is that STM is strictly a surface measurement. Part 1:

1a) *The authors can only guess the David star positions in the layers below the one they image. The ambiguity originates from the finite correlation length (~3 to 10 unit cells) of the dimerization and large number of stacking faults, as shown in the structural studies (Ritschel et al., Tanda et al.). Therefore, the extension of the CDW lattice from the region 4 to the regions 1&2 in Figure 3 is speculative and should be clearly marked as such...*

Our guess for the Star-of-David (SD) cluster positions beneath Regions 1 & 2 in Figure 2 is based on the extrapolation of the CDW lattice of the lower terraces on the right-hand-side of the image (Regions 3 & 4), extending it over to the left-hand-side. This inference is reliable given the assumption that there are no un-detected domain walls in those layers. We agree with the Referee that this assumption must be made absolutely clear to the reader. We have added a line in the main text accordingly. And we have also added a line to explain why we think the key assumption is well-founded. Please see our revision to the main text, below (the underlined part is new/revised):

"The array of white dots is extended over to the left-hand-side of the image. We make the assumption that neither of the lower layers host a domain wall hidden below the uppermost layer, so that overlaying the centres of the SD lattices of the upper terrace with this extended reference lattice shows the approximate in-plane shift between layers. We infer from the change of electronic structure between Regions 1 & 2 that the observed domain wall exists only in the

uppermost layer. (The ordinary domain walls which penetrate through multiple layers usually are not accompanied by such a change in electronic structure. See Supplementary Information.) Additional atomically resolved topographic imaging in Region 1 (see Supplementary Information) is used to determine the orientation of the SD clusters, ..."

The above revision is listed as Item 8 in the List of Changes.

Additionally, it is important to distinguish between the in-plane domain walls, the 1D defects within a given layer, and the 'stacking faults' which the Referee mentions (described by Ritschel *et al.*, and Tanda *et al.*, among others). These stacking faults are 2D planes of discontinuity only in the *vertical* stacking order. We emphasize that this disorder is a distinct issue from the in-plane domain walls and is not relevant to the analysis of Figs. 2 & 3 described in the main manuscript.

However, we now recognize it is worthwhile to give this disorder a thorough description. We add a short paragraph and the following figure (Fig. R1) in our Supplemental Information (in order to preserve the readability of the main text). This largely follows the comprehensive description given by Ritschel *et al.* [PRB **98**, 195134 (2018)]. We also add a line in our main text to direct the reader to this information, and suitable citations of recent works discussing the dimerization and disorder. These are Items 4 & 16 in the List of Changes.

Figure R1. Partial disorder in the inter-layer stacking of SD cluster bilayers. (a) Three possible sites atop which successive BLs can stack. The pair of sites circled in yellow (or blue) are symmetrically equivalent through translation by one CDW superlattice vector. The sets of red, blue and yellow-circled sites are symmetrically equivalent to each other through rotations of 120° . (b) An extended schematic showing an example of the ACAC stacking pattern with partial disorder of the T_C stacking vectors described previously [Tanda, JPSJ **53**, 476 (1984) and Ritschel, PRB **98**, 195134 (2018)]. The shifted stacking between T_A -stacked BLs switches randomly between the three types of site highlighted in (a).

1b) *The tunneling current can reach one layer beneath, as has been reported before and also stated by the authors when discussing the buried domain wall in Suppl. Fig. 1. Therefore, the spectra of the two neighboring layers would be convoluted, if no precautions are taken. I suggest the authors vary the tip-sample separation in the spectroscopy measurements to reduce the penetration depth and thus extract the clear spectra for the two terminations. I would like to see the set of spectra measured with the different tip-sample separations. I would expect the transition from clear to mixed shape (like Reg. III in Suppl. Fig. 1d) as the tip is moved closer.*

It is probably not best to think that the STM can probe one layer beneath the uppermost one. Rather, it is better to think that the STM may detect the influence, on the uppermost layer, of coupling to an underlying layer – i.e. the manifestation of the underlying layers in the measurement is *indirect* and the total signal is usually not well modelled as a linear combination or convolution of independent layer contributions. Why not? In STM, a good rule of thumb is that the tunneling probability falls off by about one order of magnitude for every Angstrom by which the tip-sample distance increases. This means that the second layer of a material with an inter-layer spacing of $\sim 6 \text{ \AA}$ yields a vanishingly small contribution to tunneling spectra. Therefore, we

shouldn't expect the density of states of the underlying layer to appear in our $dI/dV(V)$ curves in linear combination with that of the uppermost one.

Still, it is a good precaution to check the dependence of spectra on the tip-sample distance, as the Referee suggests. Figure R2 below shows that there is no significant dependence of the $I(V)$ or $dI/dV(V)$ curves on the tip-sample separation, for either of the two regularly observed surface terminations. The full detail can be found in the discussion newly added to the Supplemental Information. Please see Item 19 in the List of Changes. (Note: The sharp conductance peaks which appear in the Figure below, at the onset of the UHB at each surface, will be discussed in detail in an upcoming manuscript (in preparation).)

Figure R2. Tip-height-dependence of $dI/dV(V)$ and $I(V)$ spectra. STM topography maps acquired at surfaces of paired (a) and unpaired (b) clusters (scale bar 1 nm). Spectra were

collected in the center of SD clusters (at locations marked by the yellow dots) and sufficiently far from impurities that spectra were cluster-independent. (c)—(f) $dI/dV(V)$ spectra, and also $I(V)$ spectra, obtained at $z_{\text{set}} + z_{\text{offset}}$. (Here z_{set} is the starting (set-point) tip height using $V = 0.25$ V, $I_{\text{set}} = 1.0$ nA for (c) and $V = 0.11$ V, $I_{\text{set}} = 0.44$ nA for (d), yielding a set-point tunnel resistance $R_{\text{gap}} = 0.25$ G Ω in each case.) For the dI/dV curves in panels (a) and (b) $V_{\text{mod}} = 1$ mV, and therefore the apparent increase in the gap size as compared to those shown in Figure 1(e) of the main manuscript is probably due to the narrower energy resolution function for the lock-in detection technique. The sharp peaks in conductance at the onset of the UHB will be discussed elsewhere.

The Referee points out that there is noticeable in-gap density of states for the apparent ‘large-gap’ spectrum shown in Region III in Figure S1(d). This type of spectrum was seen in two domains found at two separate cleaved surfaces of only one 1T-TaS₂ platelet. Based on the relative rarity of this observation (found only locally and in only 2 out of 24 cleaved surfaces), and also on the tip-height dependence measurements shown above, we speculate that this in-gap density of states is not due to tip-height effects, but instead may be due to the proximity below the surface of one of the inter-layer stacking faults described above, or some other type of stacking fault.

The tip-height-dependent data (Figure R2 above) will be included in the Supplemental Information, with an explanatory paragraph. See Item 19 in the List of Changes.

2. The authors discuss Mottness collapse based on the U/t ratio. Surprisingly, the shortest inter-cluster distance is in the interlayer direction: the in-plane distance is $a\sqrt{3} \sim 1.2$ nm ($a \sim 0.7$ nm; the atomic period), the out-of-plane distance is $|a + b + c| \sim 0.83$ nm (for ACAC stacking, $c \sim 0.13$ nm; the interlayer period). It is also known, that the Mottness collapse occurs inside the domain wall (e.g. Cho et al., Skolimowski et al., Fig. 2 and Suppl. Fig. 1 in the present manuscript), where SD clusters are moved closer e.g. by one atomic period inside the plane, resulting in the distance of $3a \sim 1\text{nm} > 0.83\text{nm} \sim |a + b + c|$. Why the Mott state does not collapse in the ACAC stacking then? Careful calculations are necessary in this case, so I suggest the authors to be more careful in making implications from the simple U/t argument in such a complex multi-atom cluster with further complications like the interlayer S-S bonding etc. (cf. Lee et al.).

The Referee's point is well taken. The absolute distance between cluster sites is a necessary but not sufficient degree of freedom to model the situation. We understand that the orbital involved, located at the cluster center, is probably Ta $5d_{z^2}$ (PRB **96**, 125138 (2017), for example), with lobes pointing out-of-plane, and the inter-layer orbital overlap should be very sensitive to the inter-layer registry, as well as the distance between sites. So a complete description must make the distinction between the intra- and inter-layer hoppings t_{\parallel} and t_{\perp} . Probably the inter-layer hopping amplitude t_{\perp} is very sensitive to the lateral displacement in stacking between layers, while t_{\parallel} is not. (Domain walls might be assumed to strongly modify the local t_{\parallel} , but we do not comment further on the electronic structure of domain walls in this work.)

In light of this, we have re-written the relevant part of our discussion to frame the issue more carefully, and actually without explicitly referring to U and t parameters of the Hubbard model.

Revised version (repeated as Item 11 in the List of Changes):

"The apparent role of Mottness in the unpaired SD layer also suggests that the metallic state also observed (Region 2 in Fig. 2) is properly described as a 'Mottness-collapsed' state. The orbital localized at the SD centre is most likely Ta $5d_{z^2}$ [9,27], and due to its anisotropic projection perpendicular to the plane, it is reasonable to expect the inter-layer orbital overlap to be very sensitive to the lateral displacement between layers. In this case, upon changing the stacking from T_C to T_B , we may speculate that the inter-layer overlap increases beyond the threshold for breakdown of the Mott state."

New reference:

*[27] Yu, X.-L. et al. Electronic correlation effects and orbital density wave in the layered compound 1T-TaS₂. Phys. Rev. B **96**, 125138 (2017).*

3. Are there any evidence for a strain or SD cluster distortion in/around the smaller gap areas (e.g. Reg. II in Suppl. Fig. 1a)? Can the local stacking on the edge or near the domain wall be manipulated by strain (e.g. from the tip)? I suggest briefly commenting these questions in the text if possible.

In Figure S1(b) we show a map of in-plane displacement of the CDW for the field-of-view including both large- and small-gap domains. In this map, displacements with respect to an ideal

reference lattice are expressed in polar coordinates as color (hue for direction and saturation for magnitude). It is obtained using the so-called Lawler-Fujita scheme [Nature **466**, 347–351 (2010)], which is sensitive enough to easily detect inhomogeneity in strain of $\sim 1\%$ given a large enough field of view. Strain near the domain boundaries would be expected to show up as variations in color approaching the domain walls. Instead we see only abrupt transitions at the domain walls (on top of smooth global variations which are attributed to hysteresis in the displacement of the STM scanning piezo-tube [PRX **5**, 031022 (2015)].) Therefore, it is probably not the case that interactions between tip and sample, or the presence of the domain wall, induce significant strain.

The above information is added to the discussion of Fig. S1 in the Supplemental Information. Please see Item 17 in the List of Changes.

4. It is not clear from the present density plot of the dI/dV curves whether the gap is present very close to the step? I suggest showing the raw curves in the supplement.

We thank the Referee for this observation. Below (Figure R3), we show both the raw and normalized dI/dV curves associated with the $dI/dV(x,V)$ plot in Figure 2(b). Each curve corresponds to one pixel along the long axis of the red-tinted rectangle in Figure 2(b), but is itself the average over the column of pixels spanning the short axis. This figure makes it clearer exactly where the Mott gap breaks down near the domain wall and steps. As is consistent with previous works, we see that the variation in spectra across the domain wall occurs over a length scale of ~ 3 nm (or ~ 3 CDW lattice spacings).

A short note has been added to the existing discussion in the Supplemental Information to draw attention to the above point. See items 7 and 21 in the List of Changes.

Figure R3. Spatial dependence of conductance spectra across the step-terrace morphology. (a) Raw and (b) normalized dI/dV curves acquired in the red-tinted rectangle in Figure 2(a) of the main manuscript. The normalized data is the same as that displayed using a colormap in Figure 2(b). (Normalization is implemented by dividing the dI/dV signal by the current value at $V = -500$ mV. Curves are vertically offset by 1 nS in (a) and 3 V^{-1} in (b).) The approximate widths of the domain wall and the upper and lower steps (grey curves) are 2.9 nm, 3.3 nm, and 3.4 nm, respectively, although these are overestimates due to the fact that the line-cut is not perpendicular to the domain wall or steps.

5. *It would be better to use easily distinguishable colors for the atomic and cluster unit cells in Fig. 1b.*

The colors have been changed in the new version of Figure 1(b). The corresponding change has also been applied to the inset of Figure 1(d), and also to the similar diagrams in Figure S3. Please see item 5 in the List of Changes.

6. *I suggest the $T_C = a + b + c$ notation to be dubbed with $T_C = 2a + c$ to be consistent with a number of the previous theoretical studies.*

Our introduction to the stacking vectors $\mathbf{T}_{A,B,C}$ is now consistent with the established notation in previous literature. Specifically, we adopt the following expressions for the symmetrically distinct stacking vectors [in line with Lee *et al.*, PRL **122**, 106404 (2019)]:

$$\mathbf{T}_A = \mathbf{c},$$

$$\mathbf{T}_B = \pm \mathbf{a} + \mathbf{c},$$

$$\mathbf{T}_C = \pm 2\mathbf{a} + \mathbf{c} \text{ (or equivalently, } \mp 2\mathbf{a} \mp \mathbf{b} + \mathbf{c}\text{)}.$$

We have also added depictions of the atomic lattice vectors to Figure 1(b), accordingly. Please see item 3 in the List of Changes.

7. *I would like to bring the authors' attention to the following three papers dealing with the three-dimensional stacking: a) A. S. Ngankeu et al. Quasi-one-dimensional metallic band dispersion in the commensurate charge density wave of 1T-TaS₂. Phys. Rev. B 96, 195147 (2017). b) P. Darancet et al. Three-dimensional metallic and two-dimensional insulating behavior in octahedral tantalum dichalcogenides. Phys. Rev. B 90, 045134 (2014). c) D. Svetin et al. Three-dimensional resistivity and switching between correlated electronic states in 1T-TaS₂, Sci Rep 7, 46048 (2017)*

We thank the Referee for pointing out these papers. Citations to the papers have been incorporated into the introduction section, and also in the discussion of Mottness at the 'unpaired' surface (see Reply to Referee 2 below). Please see both Items 1 and 4 in the List of Changes.

Reply to Referee 2

The crux of the authors' argument is an unpaired surface layer hosting an odd number of electrons per unit cell which allows trivial gap opening mechanisms to be ruled out. However, this argument is not so convincing since the unpaired top layer has finite interlayer coupling to the layers underneath and can't be simply viewed as a decoupled layer. I believe it is important for the authors to show better evidence that for this stacking (and in the absence of correlation) there would indeed be a metallic surface state with an odd electron filling even though some interlayer coupling still exists. For a Mott-Hubbard gap to open this metallic surface state band has to have a small bandwidth and be half-filled, and the authors should provide more evidence that these conditions are fulfilled. Can the authors perform DFT simulations of this structure using a finite slab model to investigate the surface electronic properties? The problem here is that ARPES measurements and simulations have shown dispersion in the z-direction (PRB 96, 195147 (2017), PRB 90, 045134 (2014)) and theoretical calculation of bulk T_C -stacked TaS_2 have also exhibited metallic behavior with a ~ 400 meV bandwidth (PRL 122, 106404 (2019)). Both of these results suggest that there can be significant coupling and significant bandwidth for T_C -stacked layers, which goes against the authors' interpretation of a decoupled top layer with very narrow bandwidth (i.e., narrow enough to drive a Mott transition).

We thank the Referee for these insightful and important criticisms.

On comparison to previous ARPES and DFT works:

Various interpretations of ARPES results which appear to match well with DFT calculations based on pure T_A stacking [PRB **96**, 195147 (2017) & PRB **90**, 045134 (2014), mentioned by the Referee], pure T_C stacking [Nature Physics **11**, 328–331 (2015)], and mixed ('ACAC'-type) stacking [PRB **98**, 195134 (2018)] have all been reported. This leads to some ambiguity. Here we only note that the work mentioned by the Referee indicates a one-dimensional (out-of-plane) *metal*, definitely contrary to our observation for both types of pristine cleaved surface ('paired' and 'unpaired').

The calculations reported in PRB **90**, 045134 (2014) and PRL **122**, 106404 (2019) appear to show a large bandwidth for both pure T_A and pure T_C stacking. However, Lee *et al.*, in PRL **122**,

106404, show that the shifted-bilayer stacking (*AL* stacking in the notation of Lee *et al.*, but ‘*ACAC*’ stacking in ours) substantially reduces the band width from ~ 400 meV down to 200–300 meV, and that a gap opens in absence of U . And Lee *et al.* specifically attribute the difference to the T_C -stacking interfaces within the *ACAC* bulk structure. Therefore, it is reasonable that the inter-layer hopping between our T_C -stacked (‘unpaired’) layer and the underlying bilayer may be weak enough that it may be considered as a somewhat decoupled layer amenable to analysis in isolation.

On metallicity of the unpaired layer in the absence of correlations:

DFT calculations seem to be the only way to directly demonstrate that the system would be metallic without correlations, as we can’t escape correlation effects in the real (experimental) world. However, we can argue that if the bulk bilayer stacking can be thought of as dimerized [see arXiv:1907.11610 (2019), for example], then breaking such dimers to form the unpaired (T_C -stacked) surface leaves a uniform array of ‘dangling bonds’, which generally can be expected to form a metallic dangling bond surface state, as in the case of the metallic Si(111) 7×7 surface. Our case parallels that of the Si(111) 7×7 surface, which has odd electrons per surface unit-cell (while the bulk is an ordinary band insulator) but undergoes a transition into a Mott insulating state at ultra-low temperature [PRB **80**, 125326 (2009)]. By analogy, we may interpret the gap at the unpaired surface of $1T$ -TaS₂ as Mott localization in an otherwise metallic surface state.

We have revised and added to the discussion of these points in the main text. Although it is not possible to draw a decisive conclusion, we have made an effort to make our reasoning as clear as possible to the reader. Please see item 10 in the List of Changes.

Can the authors perform DFT simulations of this structure using a finite slab model to investigate the surface electronic properties?

The large in-plane extent of the CDW unit cell poses a serious challenge: Each SD cluster contains 39 atoms (including sulfur), and a bulk unit-cell representing the ‘*ACAC*’ or any ‘bilayer’ stacking pattern contains 78 atoms, which is still manageable. Such bilayer bulk unit-cells were investigated using state-of-the-art techniques [Nature Physics **11**, 328–331 (2015), PRB **87**, 125135 (2013), PRL **122**, 106404 (2019)]. But please note that no DFT investigations have

been reported for a slab model including surfaces and a vacuum layer, beyond only a single isolated bilayer [PRB **90**, 045134 (2014)]. This may be because a slab with a thickness of probably at least 5 bilayer unit-cells would be needed, at the bare minimum, which would already include 390 atoms. According to our correspondences with colleagues performing state-of-the-art DFT calculations, such calculations are “*not impossible*”, but are very daunting, and probably represent a substantial project in their own right. Therefore, we leave this as a challenge to be tackled in the future by computational colleagues.

Replies to additional comments:

1. *Since both bulk TaS₂ and the reported surface state are insulating, the measured STS gap size can be dramatically affected by band bending effects from the tip and other sample-related origins (as shown, for example, on Si(111) in PRB 80, 125326 (2009)). I believe this type of analysis is necessary to identify the true gap size of the unpaired layer. Also, how large is the lateral band bending effect in different domains in Fig. 2b?*

The tip-induced band-bending (TIBB) effect occurs due to the electric field from the tip penetrating into an insulating sample. The outcome is that spectroscopic features are lifted to higher energies (or sunken to lower energies) while probing unoccupied (occupied) sample states. The energy shifts monotonically follow the bias (with the same sign), but of course are zero for zero bias, so are still small while detecting a band onset quite near to E_F . We first point out that our dI/dV curves show that at both terminations of 1T-TaS₂, the onset of occupied states (lower edge of the Mott gap) is only ~10 to 20 meV below the Fermi level, so that the TIBB is very weak at the point that the onset of these occupied states is detected. Therefore, the measured energy of this lower onset should be fairly accurate.

Furthermore, when measuring the onset of the UHB, the positive bias used would be expected to induce a rigid bending of the density of states to higher energies. This would soon bring the finite density of states from -10~20 meV upwards to the Fermi level, quickly resulting in effective screening of the tip's electric field by holes. For this combination of reasons, we don't anticipate a large TIBB effect which would impact any of the conclusion drawn in this manuscript.

It is better to demonstrate this experimentally, and we have performed the suitable measurements to do this, but these measurements yield somewhat anomalous results. In the well-established framework due to Feenstra, as well as being related to the bias, the size of the

energy shift also depends on the tip-sample distance. Any shift is enhanced when the tip is close to the sample, so that for energies above E_F , any particular spectroscopic feature with energy E_0 will appear at *higher* energy E_0' when the tip-sample distance *decreases*. *i.e.* $dE_0'/dz_{\text{offset}} < 0$. In principle, this effect should be observed in the additional tip-height-dependent conductance spectra we present in Figure R2 above. Here it is helpful to zoom in to the UHB edge, as shown below in Figure R4 below, where very sharp conductance peaks (to be discussed in detail in a forthcoming manuscript) allow us to track dE_0'/dz_{offset} . We find that in fact $dE_0'/dz_{\text{offset}} > 0$, which is not amenable to interpretation in the usual TIBB model established by Feenstra.

Figure R4. Tip-height-dependence of conductance spectra. Conductance peaks at the UHB onset for paired (a) and unpaired (b) clusters, collected as described for Figure R1 above. In each case, the apex of the conductance peak, observed at the energy E_0' clearly shifts towards higher energy as the tip-sample distance increases, *i.e.* $dE_0'/dz_{\text{offset}} > 0$. This cannot be interpreted in terms of the usual TIBB model described by Feenstra [J. Vac. Sci. Technol. B **5**, 923 (1987) and Nanotechnology **18**, 044015 (2006)].

The above indicates that even if the usual band bending mechanism is in effect, another (as yet unknown) mechanism overrides it, and unfortunately the intrinsic gap size cannot be determined with greater precision at this time.

We have added a new section in our Supplementary Information to discuss these points, including the suggested citation (as well as a few more). For this, please see Item 19 in the List of Changes.

Also, how large is the lateral band bending effect in different domains in Fig. 2b?

It is well established that ordinary domain walls in 1T-TaS₂ can cause lateral band bending due to remaining charge density which we may think of as left over from partial SD clusters. It is reasonable to expect that step edges also host such remnant charge density. However, no obvious lateral band bend is seen near either of the step edges in Figure 2(b). The previously reported band bending occurred in insulating domains, whereas in Figure 2(b) shown here, the domain wall (and one of the step edges) is adjacent to a metallic domain (Region 2) where the remnant charge density from the domain wall may be allowed to diffuse away. This may explain the absence of any obvious lateral band bending around the domain wall between Regions 1 and 2, and is consistent with previous observation [Nat. Commun. **7**, 10453 (2016)].

By comparison, the usual lateral band bending, comparable with previous observation [Nat. Commun. **8**, 392 (2017)], is seen in Fig. S1(c), because the domains on both sides of each domain wall are insulating.

A new section has been added to the Supplementary Information, where the above paragraph is basically reproduced *verbatim*. For this, please see item 20 in the List of Changes.

2. Many people have measured bulk 1T-TaS₂ with STM, and it is a little surprising (and also quite interesting) to see a previously unreported cleavage plane. Did the authors do anything special to expose these cleaving planes? Is there anything special about their crystals and how is their crystal quality compared to other published results?

In our interpretation, the low temperature of sample cleavage and also the maintenance of low temperature during transfer into the STM is the important point allowing this new observation, and not the actual synthesis of the crystals themselves. The bulk stacking structure can only be accessed by surface sensitive probe if the surface is formed *after the bulk CDW structure has already set in* (and it is subsequently preserved throughout sample transfer, until measurement.) If the sample is cooled through the CDW transition temperature only *after* cleavage, a cleaved surface which retains information about the bulk stacking order may not be expected, since the presence of the surface may set boundary conditions on the CDW formation, possibly leading to a spurious result.

In our experimental procedure, the samples were cooled to $T = 77$ K (far below the transition temperature into the C-CDW phase) for cleavage and, importantly, the head of the transfer rod which was subsequently used to insert samples into the STM was cooled along with them. This ensures that the cleaved surface retains information about the pre-formed bulk stacking order.

In almost all previous STM work which show dI/dV data which can be compared against our own, the temperature of sample cleavage was either reported to be room temperature [Nat. Commun. **7**, 10956 (2016), Nat. Commun. **8**, 392 (2017), arXiv:1906.11983 (2019)], or else was unspecified [Nat. Commun. **7**, 10453 (2016), PRL **122**, 036802 (2019)]. To our knowledge, only in one work was it specified that the cleavage temperature was below the transition temperature into the C-CDW phase [PRX **7**, 041054 (2017)], but in this case we do not know whether samples were raised to $T > 180$ K during the sample transfer.

Moreover, even in our experiments, where cleavage through the pre-formed bulk charge order was achieved, the Type I spectrum was still seen for most cleaved surfaces. (Indeed, near the beginning of our own measurements, the first observation of the small gap surface (shown in Figure S1) was dismissed as an anomaly. Only later was it found to be robustly reproducible.)

We have made an effort to further emphasize the importance of the cleavage and transfer temperatures in our main text, and in our Methods section, citing the works mentioned above. Please see item 13 in the List of Changes.

List of Changes

Below, revisions are listed in order of their appearance in the manuscript. Please note that there are some revisions which are not prompted by the Referees' suggestions, but by correspondence with other colleagues, formatting considerations, the appearance of a new relevant manuscript, *etc.* The List of Changes has two parts: **Main manuscript** and **Supplemental Information**.

Main manuscript:

1) In accordance with *Nature Communications* format, the Introduction has been re-arranged. The former introductory paragraph, minus citations and abbreviations, is now the Abstract. The first paragraph of the introduction is now as follows:

“The origin of the spectral gap in many insulating materials is difficult to determine because, as well as band theoretic considerations such as the degree of band filling, electron-phonon interactions, strong electronic correlations [1,2] and other mechanisms generally can coexist and may all play some role. This is true in the decades-old charge-density-wave compound 1T-TaS₂, for which the debate over the nature of the low-temperature insulating state has only intensified in recent years [3-7]. Though the proximate cause of this insulating state is under debate, its precursor is known to be an electron-phonon driven commensurate charge-density-wave (C-CDW) phase.”

The references used above are as follows:

[1] Mott, N. F. & Peierls, R. Discussion of the paper by de Boer and Verway. Proc. Phys. Soc. Lond. **49**, 72 (1937).

[2] Imada, M., Fujimori, A. & Tokura, Y. Metal-insulator transitions. Rev. Mod. Phys. **70**, 1039 (1998).

[3] Darancet, P., Millis, A. J. & Marianetti, C. A. Three-dimensional metallic and two-dimensional insulating behavior in octahedral tantalum dichalcogenides. Phys. Rev. B **90**, 045134 (2014).

[4] Ngankeu, A. et al. Quasi-one-dimensional metallic band dispersion in the commensurate charge density wave of 1T-TaS₂ Phys. Rev. B **96**, 195147 (2017).

[5] Ritschel, T., Berger, H. & Geck, J. *Stacking-driven gap formation in layered 1T-TaS₂*. Phys. Rev. B **98**, 195134 (2018).

[6] Ligges, M. et al. Phys. Rev. Lett. **120**, 166401 (2018).

[7] Lee, S.-H., Goh, J. S. & Cho, D. Phys. Rev. Lett. **122**, 106404 (2019).

2) An additional reference has been added in support of 1T-TaS₂ as a quantum spin liquid candidate. This is now reference Z:

[16] H. Murayama et al., *Effect of quenched disorder on a quantum spin liquid state of triangular-lattice antiferromagnet 1T-TaS₂*, arXiv:1909.00583 [cond-mat.str-el] (2019).

3) The notation used to describe the possible stacking vectors has been changed:

Previous version:

“Neglecting the S atomic layers sandwiching the Ta layer, there are only three symmetrically inequivalent stacking vectors: $T_A = c$, $T_B = a + c$, and $T_C = a + b + c$ (with the latter two each having a group of symmetrical equivalents).”

Revised version:

“There are five symmetrically inequivalent stacking vectors, which may be collected into only three groups according to their length: $T_A = c$, $T_B = \pm a + c$, $T_C = \pm 2a + c$ (or equivalently, $\mp 2a \mp b + c$).”

The corresponding change has also been applied to the later mention of the planar projection of T_C when discussing the interpretation of Fig. 3c:

“... we note that single-layer steps should result in an in-plane displacement, or phase jump $\Delta\phi$, of the 2D projected CDW pattern from one terrace to the next, which should alternate between zero and non-zero (specifically $\pm 2a$, or equivalently, $\mp 2a \mp b$, the in-plane projection of T_C), as is shown in Fig. 3a.”

4) The introduction of the proposed ‘dimerized’ or ‘bilayer’ stacking structure has been expanded (new text is underlined) with additional references:

“... Ritschel et al and Lee et al recently challenged the rationale by which 1T-TaS₂ was thought to be a Mott insulator, showing that if the stacking alternates between vectors T_A and T_C as previously suggested [18–20], such that the new supercell includes two SD clusters, ab initio calculations predict an insulator without the need to invoke strong e-e

interactions [5, 7]. (It has been established that the bulk stacking structure likely alternates between T_A and a vector drawn randomly from three versions of T_C related by rotations of 120° , in a partially disordered pattern - see Supplementary Information. The ‘dimerization’ of the stacking structure into bilayers, and the disorder, have also been discussed in the interpretation of recent experimental works [21, 22].) Put simply, if the electronic unit cells contains two SDs ...”

New references:

[21] Svetin, D., Vaskivskiy, I., Brazovskii, S. & Mihailovic, D. Three-dimensional resistivity and switching between correlated electronic states in 1T-TaS₂. Sci. Rep. 7, 46048 (2017).

[22] Stahl, Q. et al., Collapse of layer dimerization in the photo-induced hidden state of 1T-TaS₂. arXiv:1907.11610 (2019).

5) Figure 1 has been revised to use clearer color contrasts for the unit cells, and to explicitly depict the atomic lattice vectors, **a** and **b**, for un-distorted 1T-TaS₂. Also, the maximum and minimum topographic heights are included for both the maps in panel (d).

Figure R5. The revised version of Fig. 1.

6) The maximum and minimum topographic heights have been added to all the figures. See for example Fig. R5 above.

7) Relating to Figure 2(b), a note has been added in the caption to direct readers to the spatially resolved raw dI/dV and normalized dI/dV curves now shown in the Supplemental Information.

8) Assumptions used in the interpretation of Figure 3 are made clear (new text is underlined):

“The array of white dots is extended over to the left-hand-side of the image. We make the assumption that neither of the lower layers host a domain wall hidden below the uppermost layer, so that overlaying the centres of the SD lattices of the upper terrace with this extended reference lattice shows the approximate in-plane shift between layers. We infer from the change of electronic structure between Regions 1 & 2 that the observed domain wall exists only in the uppermost layer. (The ordinary (intrinsic) domain walls which penetrate through multiple layers usually are not accompanied by such a change in electronic structure. See Supplementary Information.) Additional atomically resolved topographic imaging in Region 1 (shown in the Supplementary Information) is used to determine the orientation of the SD clusters, depicted in the zoom-in images of Figs. 3c,d. The in-plane components of the stacking vectors for Regions 1 & 2 are then discernible.”

9) A small modification has been made to the line mentioning relative formation energies for the two surfaces (new words are underlined):

“The fact that this surface was the most common outcome from cleavage (18 out of 24) indicates the energetic favourability of cleaving between BLs, rather than through them, suggesting non-negligible intra-BL bonding, or dimerisation.”

10) The text discussing the spectral gap of the ‘unpaired’ surface has been extensively revised:

“The surface of unpaired SD clusters (Regions 1 & 4) represents a new and perhaps qualitatively distinct system, which may allow us to disentangle the role of strong e-e interactions from that of unit-cell doubling in the electronic structure. First, we note that if the BL-stacked bulk charge order can be considered as dimerised, then breaking the dimerisation by terminating the structure with a layer of unpaired clusters may ordinarily be expected to leave a metallic ‘dangling bond’ surface state, as is the case at the well-known Si(111)-7×7 surface.

As opposed to previously suggested structure of purely T_A -stacked SD cluster, which yields a large out-of-plane bandwidth and a one-dimensional out-of-plane metal [3,4], the ACAC stacking pattern realised here has been predicted to result in suppression of the out-of-plane bandwidth which can be attributed specifically to the T_C stacking interfaces [7]. In this case, the T_C -stacked, unpaired surface layer here may be sufficiently decoupled from the underlying BLs that the usual justification for attributing the gap to Mott localisation can be recovered: It is a system with an odd number of electrons per (surface) unit cell, and yet which is insulating. This mirrors the aforementioned Si surface, which also has an odd number of electrons per surface unit cell but undergoes a transition to a Mott insulating state at very low temperature [26]. Hence, the spectral gap observed in the unpaired SD layer, where an otherwise metallic surface state is expected, may be attributed to Mott localisation.”

New reference:

*[26] Modesti, S., Gutzmann, S. H., Wiebe, J. & Wiesendanger, R. Correction of systematic errors in scanning tunneling spectra on semiconductor surfaces: The energy gap of Si(111)-7×7 at 0.3 K. Phys. Rev. B **80**, 125326 (2009).*

11) The following text has been added to the discussion of metallicity in Region 2 of Figs. 2 & 3 (new text is underlined):

“The apparent role of Mottness in the unpaired SD layer also suggests that the metallic state also observed (Region 2 in Fig. 2) is properly described as a ‘Mottness-collapsed’ state. The orbital localized at the SD centre is most likely a Ta $5d_{z^2}$ [9,27], and due to its anisotropic projection perpendicular to the plane, it is reasonable to expect the inter-layer orbital overlap to be very sensitive to the lateral displacement between layers. In this case, upon changing the stacking from T_C to T_B , we may speculate that the inter-layer overlap increases beyond the threshold for breakdown of the Mott state.”

New reference:

[27] Yu, X.-L. et al. *Electronic correlation effects and orbital density wave in the layered compound 1T-TaS₂*. Phys. Rev. B **96**, 125138 (2017).

12) We found through discussions with other colleagues that the schematic in Figure. 3(e) showing the CDW stacking configurations for the three observed surfaces needs to be easier to understand. A schematic that offers 1:1 comparison with the data in Figs. 2(b) & 3(b) is now shown. This schematic should also compare well against the one shown for the expected pristine step-terrace formation shown in (a). As below:

Figure R6. The new version of the ‘visual summary’ in Figure 3(e).

13) Concerning how cleavage and sample transfer temperatures enable our new observations:

In the main text (the modified part is underlined):

“In this work, samples were cleaved, transferred to the STM and measured at temperatures far below the transition temperature at which the C-CDW sets in (i.e. far below ~180 K, see Methods), and the bulk structure of the CDW should be preserved such that measurements on a large number of cleaved surfaces may show evidence of the ACAC pattern.”

In the Methods section (new text is underlined):

“... Samples were cleaved in ultra-high vacuum ($\sim 10^{-10}$ Torr) at 77 K. Importantly, the head of the transfer rod which was subsequently used to insert samples into the pre-cooled STM was cooled along with each sample to a temperature near 77 K, and samples were inserted within a few tens of seconds after cleavage. We suggest that the

maintenance of temperature far below the onset of the C-CDW phase (~180 K) is important for achieving cleaved surfaces which yield information about the pre-formed bulk stacking structure. For samples cooled through the C-CDW transition only after cleavage, a cleaved surface which retains information about the bulk stacking order may not be expected, since the presence of the surface may set boundary conditions on the CDW formation. In most previous STM reports which show dI/dV data which can be compared against those presented here, the temperature of cleavage was either reported to be room temperature [24, 25, 34], or else was unspecified [23, 35]. To our knowledge, only in one work was it specified that the cleavage temperature was below the transition temperature into the C-CDW phase [9], and there a spectrum was shown which appears very similar to the the Type 1 spectrum presented in the current work (Fig. 1e). Moreover, in the present work, where a bulk-like termination of the 3D charge order was achieved, the Type 1 spectrum was still seen for most cleaved surfaces. Thus, our results are not inconsistent with previous reports.

The STM used was a Unisoku 1300 low temperature system, of which the STM head has been replaced with a homemade one [36]. ...

New reference:

[34] Aishwarya, A. et al. Visualizing 1D zigzag Wigner crystallization at domain walls in the Mott insulator TaS₂. arXiv:1906.11983 [cond-mat.str-el] (2019).

14) While making Figure R3 above, a small error was found: The dI/dV(x,V) data shown in Figure 2(b) was normalized according to the I(V) value at V = -500 mV, not +500 mV. The description in the Methods section has been revised accordingly. This difference does not have any effect on the conclusions drawn from the data. New text is underlined below:

“The dI/dV spectroscopy data were normalized according to the I(V) value at V = -500 mV, for Fig. 2b, and at V = +500 mV elsewhere. This somewhat compensates for the large difference in raw signal intensity caused by differing tip-sample distances while scanning at V = 250 mV over areas with large or small spectral gaps, and is performed only to aid visualization. The raw dI/dV curves for Fig. 2b are shown in the Supplemental Information.”

15) We have made an addition to the Acknowledgments section (new entry is underlined):

“We are grateful to Y. Kohsaka, T. Machida, J. Lee and P. A. Lee for helpful discussions.”

Additions to the Supplemental Information:

16) A new section is added to the Supplemental Information, including Figure R1 (on page 2 of this Reply Letter), the accompanying caption, and the following paragraph:

“The depiction of the three-dimensional pattern of charge order shown in Fig. 1c is not comprehensive. The stacking vector T_C has three symmetrically equivalent instantiations related to each by rotations of 120° (for example, $2\mathbf{a} + \mathbf{b}$, $-\mathbf{a} + \mathbf{b}$, and $-\mathbf{a} - 2\mathbf{b}$), and it has been established that the extended stacking pattern features a partial disorder in which T_C varies randomly between these three. The coherence length for ordered stacking within this partially disordered pattern is ~ 3 to 10 unit-cells [1-4]. A depiction of the extended stacking pattern is shown in Fig. S1 below. In Figs. 2 & 3 of the main work, we observe only a thin slice of the perpendicular stacking, and so we cannot comment on the degree of order or disorder, or the coherence length. This would require simultaneous observation of 10~100 steps and terraces – not experimentally feasible with current techniques.”

New references:

*[1] Nakanishi, K. & Shiba, H. Theory of Three-Dimensional Orderings of Charge-Density Waves in $1T\text{-TaX}_2$ (X: S, Se). *J. Phys. Soc. Jpn.* **53**, 1103–1113 (1984).*

*[3] Ishiguro, T. & Sato, H. Electron microscopy of phase transformations in $1T\text{-TaS}_2$. *Phys. Rev. B* **44**, 2046–2060 (1991).*

17) A short note about lateral strain (or the absence thereof) near the domain boundary has been added to the Supplemental Information, in the discussion for Figure S2:

“As an aside, lateral strain near the domain boundaries would be expected to show up as variations in color approaching the domain walls. Instead we see only sharp transitions at the domain walls, on top of smooth global variations which are attributed to hysteresis in the displacement of the STM scanning piezo-tube [6]. Therefore, it is probably not the case that interactions between tip and sample induce significant lateral strain near the domain walls.”

New reference:

[6] Watashige, T. et al., Evidence for Time-Reversal Symmetry Breaking of the Superconducting State near Twin-Boundary Interfaces in FeSe Revealed by Scanning Tunneling Spectroscopy. Phys. Rev. X **5**, 031022 (2015)

18) Through correspondence with other colleagues since the time of submission, we found that it will be helpful to include the topographic line profile across the step-terrace morphology shown in Figs. 2 & 3. The different apparent step heights between the various surface terminations are shown, and a possible origin of the difference is described:

“C. Supplementary data for Fig. 2.

Figure S4 below shows the topographic line profile across the step-terrace morphology examined in the main work. The height of each step corresponds closely to the expected inter-layer spacing for 1T-TaS₂ ($c \approx 6 \text{ \AA}$). The step from the large gap terrace (Region 3) up to the metallic and small gap terraces (Regions 1 and 2) is larger than from the lower small gap terrace (Region 4) up to the large gap terrace (Region 3). A likely reason for this is the higher integrated density of states between eV and zero for the small gap spectrum, resulting in a greater tip-sample distance at these set-point parameters ($V = 0.25 \text{ V}$, $I_{\text{set}} = 0.125 \text{ nA}$).

Figure S4. Topographic step heights of the step-terrace morphology. a The same STM topograph shown in Fig. 2a of the main manuscript. **b** Apparent topographic height plotted along the long axis of the red-tinted rectangle shown in a (averaged over the short axis of the rectangle.)”

19) The tip-height-dependent $I(V)$ and $dI/dV(V)$ results shown above in Figure R2 (and caption) have been added to the Supplemental Information, with an explanatory paragraph. Here we also comment on the best estimate of the actual gap sizes for each surfaces:

“E. Tip- and DW-induced band bending effects.

For each of the two regular types of 1T-TaS₂ surface, the dependence of the conductance spectrum on the tip-sample separation was measured, as shown in Fig. S7. Although the absolute tip-sample separation generally cannot be known in order to compare one measurement to another, a rough proxy for it, which can be used for comparison of samples with like spectroscopic character, is the set-point tunneling gap resistance $R_{\text{gap}} = V/\Lambda_{\text{set}}$. For the lowest tip-sample distance presented in Fig. S7 (darkest curves), $R_{\text{gap}} = 0.25 \text{ G}\Omega$, which is significantly lower (tip closer to sample) than for the measurements shown elsewhere in this work ($R_{\text{gap}} = 2.0 \text{ G}\Omega$ for Fig. 2(a) and (b), for example). From this we can safely conclude that we do not see a simple linear combination of two different DOS spectra, or that one type smoothly transforms into the other with varying tip height. (The sharp conductance peaks which appear in the Figure below, at the onset of the UHB at each surface, will be discussed in detail in the future (manuscript in preparation).)

The slightly larger apparent gap size in the conductance spectra here, as compared to those shown in Fig. 1(a) of the main manuscript, is probably due to the significantly smaller lock-in modulation amplitude used here ($V_{\text{mod}} = 1 \text{ mV}$ here, as opposed to 10 mV elsewhere in this work).

As both of the regularly observed surfaces (Types 1 & 2) of 1T-TaS₂ are insulating, the observed spectral gap sizes may be influenced by tip-induced band-bending (TIBB) artifacts. The TIBB effect occurs due to the electric field from the STM tip penetrating into an insulating sample. The outcome is that spectroscopic features are lifted to higher energies (or sunken to lower energies) while probing unoccupied (occupied) sample states. The energy shifts of spectral features in the sample monotonically follow the relative energy of the tip (with the same sign), so are still small while detecting a band onset quite near to E_F . We first point out that our dI/dV curves show that at both terminations of 1T-TaS₂, the onset of occupied states (lower edge of the Mott gap) is only $10\sim 20 \text{ meV}$ below the Fermi level, so that the TIBB is very weak at the point that

the onset of these occupied states is detected. Therefore, the measured energy of this lower onset should be fairly accurate.

Furthermore, when measuring the onset of the UHB, the positive bias used would be expected to induce a rigid bending of the density of states to higher energies. This would soon bring the finite density of states from -10~20 meV upwards to the Fermi level, quickly resulting in effective screening of the tip's electric field by holes. For this combination of reasons, we don't anticipate a large TIBB effect which would impact any of the conclusion drawn in this manuscript.

The absence of strong tip-height-dependent TIBB energy shifts of the upper and lower band onsets indicates that these measurements reflect reasonably well the intrinsic gap sizes. The current spectra are not amenable to detailed correction using previously established methods such as that formulated by Feenstra et al. [7,8]. In Feenstra's framework, it is predicted that in a semiconductor or insulator, a given unoccupied state ordinarily residing at E_0 should be observed at an energy E_0' which shifts increasingly higher as the tip-sample separation is decreased – i.e. $dE_0'/dz_{\text{offset}} < 0$. Looking in detail at Fig. S7, we see that in fact $dE_0'/dz_{\text{offset}} > 0$ for the observed onsets of unoccupied states at both surfaces. To our knowledge, no model of tip-induced band bending yet describes this behaviour. Therefore, the apparent gaps measured here represent the best available estimates of the actual gap sizes.”

New references:

*[7] Feenstra, R. M. & Stroscio, J. A. Tunneling spectroscopy of the GaAs(110) surface. J. Vac. Sci. Technol. **5**, 923 (1987).*

*[8] Feenstra, R. M., Dong, Y., Semtsiv, M. P. & Masselink, W. T. Influence of tip-induced band bending on tunnelling spectra of semiconductor surfaces. Nanotechnology **18**, 044015 (2006).*

20) In a new section, the following paragraph has been added to discuss the lateral band bending near the domain walls and step edges:

“It is well established that ordinary domain walls in 1T-TaS₂ can cause lateral band bending due to remaining charge density left over from partial SD clusters. It is reasonable to expect that step edges also host such remnant charge density. However,

no obvious lateral band bend is seen near either of the step edges in Figure 2(b). The previously reported band bending occurred in insulating domains, whereas in Figure 2(b) shown here, the domain wall (and one of the step edges) is adjacent to the metallic domain (Region 2) into the remnant charge density may be allowed to diffuse away. This may explain the absence of any obvious lateral band bending around the domain wall between Regions 1 and 2, and is consistent with previous observation [9]. By comparison, the usual lateral band bending, comparable with previous observation [10], is seen in Fig. S1(c), because the domains on both sides of each domain wall are insulating.”

References:

*[9] Cho, D. et al. Nanoscale manipulation of the Mott insulating state coupled to charge order in 1T-TaS₂. Nat. Commun. **7**, 10453 (2016).*

*[10] Cho, D. et al. Correlated electronic states at domain walls of a Mott-charge-density-wave insulator 1T-TaS₂. Nat. Commun. **8**, 392 (2017).*

21) Figure R3 above, showing the spatially resolved raw and normalized dI/dV curves, has been added to the Supplemental Information, and a suitable note has been added to the main manuscript text to direct the reader to it (In the caption of Fig. 2).

22) The manuscript now includes a Data Availability statement:

“The data that support the findings presented here are available from the corresponding author upon reasonable request.”

Reviewers' comments:

Reviewer #1 (Remarks to the Author):

Unfortunately, the self-consistent understanding of the origin of the gap and the spectral shape of the key object of this paper – the unpaired layer – is still missing. I believe the spectral shape and its origin have to be elucidated prior to accepting the paper. Otherwise, I appreciate the authors' efforts in improving the paper and stand to my previous assessment of the importance the present results.

1. Origin of the spectrum of the unpaired layer in the simple case (T_C stacking).

I am puzzled by the presence of additional peaks in the spectrum of the unpaired layer, compared to the well-understood spectrum of the paired one.

My naïve guess was that there is some kind of convolution of the spectra of the underlying paired layer and that of the unpaired – this would explain the additional peaks. However, the distance dependence presented by the authors in their reply shows that such simple picture is inapplicable.

The present authors' argument – the suppressed out-of-plane bandwidth – in the first approximation seems to account for the gap size only, not the spectral shape. However, the spectral shape change is the qualitative effect. It is very nicely seen in the waterfall plot of the spectra in Fig. S5b if one extends the position of lower and upper Hubbard bands from Region 3 (paired layer) to the other regions (see Fig. RR1 attached).

Below are my requests/questions (here I will use Figure 1e for clarity):

1a. Please, assign the peaks in the red and blue curves to Hubbard bands and CDW bands respectively (as it was done in e.g. NatComm 7, 10956 (2016) or NatComm 7, 10453 (2016)).

1b. Please, discuss the qualitative change in the spectral shape mentioned above.

1c. The spectrum of the unpaired layer seems to have two split Hubbard bands (e.g. the two peaks around -200meV in the red curve are symmetrically shifted with respect to the original LHB in the blue curve). Could it be that the Hubbard bands are split due to the two inequivalent hopping integrals that appear due to the asymmetry of the David star position in the unpaired layer with respect to those in the underlying paired layer?

1d. What happens to the CDW bands in the unpaired layer?

2. Origin of the spectrum of the unpaired layer in the complex case (T_B stacking).

In the supplemental Figure S2e the authors show the possible arrangements of the star-of-David clusters in different regions around the top layer and buried domain walls. I disagree with their assignment of the shifts in the top layer. According to their construction, the SD move further from each other inside the top domain wall (marked blue dashed line). However, they seem to move closer in the experimental image (see constructions in Fig. RR2 and RR3 attached). I assume here that all the shifts in the domain walls are less (by absolute value) than 2 atomic periods, following the discussions in NatComm 7, 10956 (2016), NatMatRev 2, 17004 (2017) and npj QuantMat 4, 32 (2019).

Question 2a: Once and if the shift is corrected, how the region II in Fig. S2 is different structurally and spectrally from the region 2 in Figs. 1, 2 in the main text?

Question 2b: Given the above, is it possible to prove the following statement you make in the updated manuscript: "The orbital localized at the SD centre is most likely Ta 5dz₂ [9,27], and due to its anisotropic projection perpendicular to the plane, it is reasonable to expect the inter-layer orbital overlap to be very sensitive to the lateral displacement between layers. In this case, upon changing the stacking from TC to TB, we may speculate that the inter-layer overlap increases beyond the threshold for breakdown of the Mott state."?

Technical comment: in the supplementary section B the authors refer to the non-existent figures S1a and S1e, but probably mean S2a and S2e.

FIG. RR1

From Fig. 1

e
From Fig. S2

b
From Fig. S5

FIG. RR2

See the yellow and white arrows construction

FIG. RR3

See the yellow and red
arrows construction

Reviewer #2 (Remarks to the Author):

In the revised version of their paper, Butler et al. have included a significant amount of supplementary data and made changes throughout the main text. The experimental data is excellent and convinces me that this work should be published in Nature Communications, but the lack of theoretical support is still the weak point of this paper. I would recommend publication so long as the authors can address the following questions.

1. I agree with the authors that DFT with a 5 double-layer system would be quite a bit of work, and is perhaps too much to ask for in this manuscript. On the other hand, I feel that the arguments provided by the authors are still somewhat unconvincing to establish the metallic surface state without correlation. (One can easily come up with counter-arguments to the "dangling bonds" surface layer, for example, that the Tc-Ta-Tc-Ta structure can also be viewed as a perfect stacking of Tc-stacked bilayers.) Perhaps the authors should consider performing tight-binding modelling regarding the surface state. My intuition is that this alternating Ta-Tc stacking of the cluster SD orbital is very similar to the Su-Schrieffer-Heeger model (along the out-of-plane direction), where the hopping is smaller for the Tc stacking and larger for the Ta stacking. When the sample is cleaved across the bilayer, one breaks the stronger bond, and in the spirit of SSH, should get an edge/surface state (even in the presence of a finite weak bond strength) that lies in the gap (although in this case I suppose it would be a 2D band in the gap) (this is the "topologically nontrivial" case). And the other cleaving plane should not yield a surface state (i.e., the "topologically trivial" case). This can be verified more rigorously by solving a 3D tight-binding model using only the cluster SD orbital with in-plane hoppings, out-of-plane Ta hopping, and out-of-plane Tc hopping. I think this would be very straightforward and would greatly strengthen the authors' story. I have seen related arguments presented in talks by Sung-Hoon Lee (I'm not sure if it is published).

2. Regarding the band bending effects, seeing the gap size increase with increased current seems quite normal. Others have seen this in other transition metal dichalcogenides. The idea here is that the ratio of the voltage drop between the first layer and second layer compared to the drop between tip and first layer increases as the tip gets closer. (This gets a bit complicated when the tip and sample have a large work function mismatch, where the contact potential dominates and VB and CB can, for example, bend in the same direction. I believe Feenstra's simulations used a large contact potential.) The discussion in the supplement should be modified to reflect this.

Reply Letter (round 2)

We are very grateful to both the Referees for their careful reading of our revised manuscript, and their valuable comments, which have again helped us to substantially improve it. Below we describe our revisions in response, including additional experimental results. As in the previous reply, we reproduce some text from the Referees' remarks and highlight these in red, and where we make additions or revisions to the text of the manuscript, those are highlighted in blue. The figure numbers continue on from the previous Letter (i.e. starting from Fig. R7).

This Reply Letter is followed by a comprehensive List of Changes to the manuscript, where the revisions and additions are listed in order of their appearance in the manuscript and Supplementary Information.

Reply to Referee 1

Reviewer #1 (Remarks to the Author):

Unfortunately, the self-consistent understanding of the origin of the gap and the spectral shape of the key object of this paper 'the unpaired layer' is still missing. I believe the spectral shape and its origin have to be elucidated prior to accepting the paper. Otherwise, I appreciate the authors' efforts in improving the paper and stand to my previous assessment of the importance the present results.

1. Origin of the spectrum of the unpaired layer in the simple case (T_C stacking). I am puzzled by the presence of additional peaks in the spectrum of the unpaired layer, compared to the well-understood spectrum of the paired one.

[...]

The present authors' argument 'the suppressed out-of-plane bandwidth' in the first approximation seems to account for the gap size only, not the spectral shape. However, the spectral shape change is the qualitative effect. It is very nicely seen in the waterfall plot of the spectra in Fig. S5b if one extends the position of lower and upper Hubbard bands from Region 3 (paired layer) to the other regions (see Fig. RR1 attached).

Below are my requests/questions (here I will use Figure 1e for clarity):

1a. Please, assign the peaks in the red and blue curves to Hubbard bands and CDW bands respectively (as it was done in e.g. NatComm 7, 10956 (2016) or NatComm 7, 10453 (2016)).

1b. Please, discuss the qualitative change in the spectral shape mentioned above.

Please allow us to answer these two points together in the discussion below. In answer to the Referee's comment regarding the origins of particular spectral features, we are reluctant to definitively assign a particular peak to the CDW, even for the case of the paired cluster surface. Previous reports, which appear to measure the surface which we have demonstrated is the paired surface, have ventured to identify certain spectral peaks with the CDW, but these have appeared variously at around -375 meV [Ma *et al.*, Nat. Comms **7**, 10956 (2016)], around -450 meV [Cho *et al.*, Nat. Comms **7**, 10453 (2016)], and even as low as -650 meV [Zhu *et al.*, PRL **123**, 206405 (2019)]. Therefore, even the commonly observed paired surface is not as well understood as it may seem from any one paper, and more generally, STM observations alone are probably not well suited to definitively argue the underlying mechanism behind such features.

We are prepared to provide a more complete phenomenological description of the spectral feature at each surface, using newly included data below. But please note that the curves shown in Figure 1e of the main manuscript are not very suitable for the purpose of identifying or even describing all the spectral features, as those spectra were acquired very near the centers of particular SD clusters. We hope the relevance of this will become clear below.

Fig R7. Spatial distributions of the spectral features at the paired and unpaired surfaces.

(a) dI/dV spectra collected at a Type 1 surface, at the center of a typical SD cluster (light blue curve), and at the point between clusters (dark blue curve), as marked by the light and dark blue dots in the topography map. The spatial distributions of spectral features at four representative energies (marked by vertical dashed lines on the spectra) are shown in the dI/dV maps underneath. (b) The corresponding spectra for the Type 2 surface, acquired at the locations marked by the light and dark red dots in the respective topography image. The spatial distributions of spectral features at six representative energies are shown in the dI/dV maps. At the Type 2 surface, the first two features below E_F both appear to be localized at the cluster centers. Here, spectroscopic imaging was performed using the set-point parameters $V = -500$ mV, $I = 0.5$ nA, and $V_{\text{mod}} = 10$ mV. Scale bar 1 nm.

In Figure R7 above, we elucidate the spatial distributions of the various spectral features, which may help to understand how each one corresponds to the Mott-localized orbitals and the valence and conduction bands (VB and CB). We do not attribute any feature specifically to the CDW, as mentioned above. The upper panel shows that the peaks either side of the gap for the paired surface are localized at the cluster centers, and the CB and VB appear as honeycomb-like patterns around the cluster peripheries. This is all consistent with the nice spatially resolved data shown previously by Qiao *et al.*, in PRX **7**, 041054 (2017).

As can be seen in the lower panel, both the peaks just below the Mott gap for the unpaired surface (at around -120 meV and -240 meV) are localized at the cluster centers, so the lower lying of the two peaks should *not* be associated with the CDW formation. The reason is that orbitals which reconstruct upon formation of the SD superstructure are thought to be the twelve which lie around the periphery of each cluster, similar to the image at -450 meV. (In the work of Qiao *et al.*, this feature is conservatively attributed to the valence band, not specifically to the CDW, and we follow in the same spirit.) From the localization of the two peaks at the cluster centers, it is possible that these peaks can be described as a ‘split LHB’ as the Referee suggests (and likewise for the UHB). However, from the observations available to us we are not able to pin down why the LHB (or UHB) may split. Please note that the spatial distributions of the CB and VB features are very similar for both surfaces, indicating that they probably reflect stacking-independent electronic structures.

The above figure and a suitable description have now been added to our revised Supplementary Information (Item 5 in the List of Changes).

1c. The spectrum of the unpaired layer seems to have two split Hubbard bands (e.g. the two peaks around -200meV in the red curve are symmetrically shifted with respect to the original LHB in the blue curve). Could it be that the Hubbard bands are split due to the two inequivalent hopping integrals that appear due to the asymmetry of the David star position in the unpaired layer with respect to those in the underlying paired layer?

Please note that a T_C -stacked SD cluster actually has three underlying ‘nearest neighbors’ (for a very loose definition of ‘nearest neighbors’). The three sites below all have slightly different absolute distances. However, the T_C -stacked site is very close to the point which is equidistant to the three neighbors below (just slightly more than half of one Ta-Ta lattice constant away). Therefore, all three of the interlayer hopping integrals across the T_C interface are probably very

similar, and we do not expect the difference between them to be the cause of the apparent additional peak in the conductance spectrum.

1d. What happens to the CDW bands in the unpaired layer?

As shown in Fig. R7 above, the spatial distributions of the CB and VB features (imaged at +400 meV and -450 meV, respectively) are very similar for both surfaces. This is also a good indication that the in-plane behavior of CDW-related bands is not affected by the inter-layers stacking.

To conclude, within the scope of this manuscript, we contend that the important observations are, in brief, (i) that we have experimentally established that the alternating ‘ACAC’ stacking (‘unit-cell doubling’ in the title) is realized in 1T-TaS₂, and that (ii) despite this unit cell doubling, we still see signs of a correlation-driven gap at the surface where the doubling is broken. (A more thorough argument for this has now been added. Please see our reply to the second Referee.) The detailed mechanisms determining the differing spectral shapes at each surface are certainly interesting and deserve investigation, but are beyond the intended scope of this manuscript, and beyond the suitability of the tools used here to elucidate confidently.

2. Origin of the spectrum of the unpaired layer in the complex case (T_B stacking). In the supplemental Figure S2e the authors show the possible arrangements of the star-of-David clusters in different regions around the top layer and buried domain walls. I disagree with their assignment of the shifts in the top layer. According to their construction, the SD move further from each other inside the top domain wall (marked blue dashed line). However, they seem to move closer in the experimental image (see constructions in Fig. RR2 and RR3 attached). I assume here that all the shifts in the domain walls are less (by absolute value) than 2 atomic periods, following the discussions in NatComm 7, 10956 (2016), NatMatRev 2, 17004 (2017) and npj QuantMat 4, 32 (2019).

We thank the Referee for the important observation that our scenario for the domain wall configuration doesn’t match the accompanying data. This has been corrected. As an aside, we point out that it is not yet well established whether domain walls necessarily have the effect of pushing SD clusters at either side closer towards each other. It seems that Ma *et al.* [Nat.

Comms **7**, 10956 (2016)] have claimed to observe one of the possible domain wall configurations which instead pull the SD cluster further apart. In producing the revised figure, we found that there are multiple configurations which will self-consistently satisfy the general criteria for both domain walls simultaneously, and also match our data. Nevertheless, we have chosen a configuration which pushes the clusters closer together by only one lattice spacing, as that seems to be more commonly observed [Nat. Comms **7**, 10453 (2016), PRL **122**, 036802 (2019)]. Below is the new version of Fig S2e:

Figure R8. The new version of Figure S2e. The in plane shifts associated with the domain walls have been corrected to be consistent with more commonly observed cases in previous publications, and more importantly, with the data shown in Fig. S2a. Note that the shifts in Regions I and II are in opposite directions.

Question 2a: Once and if the shift is corrected, how the region II in Fig. S2 is different structurally and spectrally from the region 2 in Figs. 1, 2 in the main text? Question 2b: Given the above, is it possible to prove the following statement you make in the updated manuscript: “The orbital localized at the SD centre is most likely $T_a 5d_{z^2}$ [9,27], and due to its anisotropic projection perpendicular to the plane, it is reasonable to expect the inter-layer orbital overlap to be very sensitive to the lateral displacement between layers. In this case, upon changing the stacking from TC to TB, we may speculate that the inter-layer overlap increases beyond the threshold for breakdown of the Mott state.”?

In answer to the Referee’s question, we hope that it is now clear that Region II as depicted above and in Fig. S2e can be self-consistently interpreted as a T_C -stacked ‘unpaired’ layer,

similar to Regions 1 and 4 in Fig. 2 of the main manuscript, and with a similar ‘small gap’ conductance spectrum. It is both structurally and spectrally distinct from Region 2 of Fig. 2 of the main manuscript, which is a T_B -stacked unpaired layer, and is metallic (possibly due to Mottness-collapse driven by enhanced orbital overlap, in our interpretation).

Technical comment: in the supplementary section B the authors refer to the non-existent figures S1a and S1e, but probably mean S2a and S2e.

We thank the Referee again for all the above comments, and especially for the very careful reading of the manuscript. This error has been corrected in the revised version.

Reply to Referee 2

Reviewer #2 (Remarks to the Author):

In the revised version of their paper, Butler et al. have included a significant amount of supplementary data and made changes throughout the main text. The experimental data is excellent and convinces me that this work should be published in Nature Communications, but the lack of theoretical support is still the weak point of this paper. I would recommend publication so long as the authors can address the following questions.

1. I agree with the authors that DFT with a 5 double-layer system would be quite a bit of work, and is perhaps too much to ask for in this manuscript. On the other hand, I feel that the arguments provided by the authors are still somewhat unconvincing to establish the metallic surface state without correlation. (One can easily come up with counter-arguments to the “dangling bonds” surface layer, for example, that the Tc-Ta-Tc-Ta structure can also be viewed as a perfect stacking of Tc-stacked bilayers.) Perhaps the authors should consider performing tight-binding modelling regarding the surface state. My intuition is that this alternating Ta-Tc stacking of the cluster SD orbital is very similar to the Su-Schrieffer-Heeger model (along the out-of-plane direction), where the hopping is smaller for the Tc stacking and larger for the Ta

stacking. When the sample is cleaved across the bilayer, one breaks the stronger bond, and in the spirit of SSH, should get an edge/surface state (even in the presence of a finite weak bond strength) that lies in the gap (although in this case I suppose it would be a 2D band in the gap) (this is the “topologically nontrivial” case). And the other cleaving plane should not yield a surface state (i.e., the “topologically trivial” case). This can be verified more rigorously by solving a 3D tight-binding model using only the cluster SD orbital with in-plane hoppings, out-of-plane T_a hopping, and out-of-plane T_c hopping. I think this would be very straightforward and would greatly strengthen the authors’ story. I have seen related arguments presented in talks by Sung-Hoon Lee (I’m not sure if it is published).

The Referee suggests that it is easy to come up with counter-arguments to the expectation of a metallic state at the unpaired surface in the absence of correlations. However, the example the Referee gives, proposing that each set of T_C -stacked layers could also be considered as the appropriate even-electron cell (the Peierls-like dimer), does not seem applicable, for reasons we would like to explain below.

In our previous Reply Letter, we drew attention to the *ab initio* calculations performed by Lee *et al.*, which showed that the T_C -stacked interface, specifically, should cause sufficient suppression of out-of-plane bandwidth to open a gap overall for the ‘ACAC’ structure. This matches well with the basic intuitions that (i) the larger absolute cluster-cluster distance across the T_C -stacked interface should lead to much smaller overlap, and (ii) the lateral offset across the T_C -stacked interface should also lead to much smaller overlap due to the d_{z^2} character of the relevant orbitals. Given these reasons, together with the results of Lee *et al.*, it is very implausible that the Peierls dimers are pairs of clusters straddling a T_C interface, instead of the on-top-stacked (T_A) pairs, as we suggest.

Additionally, on this point: even if we imagine that the dimer is actually formed across the T_C interface, our other observed surface (T_A -stacked, and with the larger gap) would then represent the ‘unpaired’ layer resulting from the broken dimer. As *both* surfaces are seen to be gapped, the alternative choice of dimer arrangement doesn’t allow one to escape the conclusion that electronic correlations must open the gap at (at least) one of the two surfaces.

Still, we take the Referee’s point that we need to affirmatively explain why, in the absence of correlations, we should normally expect a metallic state at the unpaired (T_C) surface, and after discussions with our theorist colleagues we have added such a thorough and comprehensive

argument, described below. It is in line with the one described for the SSH model by the Referee, but we show its validity for a system extended in 3D (i.e. stacked 2D layers with a termination facing the vacuum). Please note that after discussions with our colleagues, we have determined that performing tight-binding calculations would not meaningfully add to the argument as presented below, as the important behavior of the 3D model in question can be understood easily enough without demonstrating it explicitly through calculations. Also, it is valid for the range of relevant in-plane and out-of-plane hopping strengths, beyond any specific choice we may make for a given calculation.

The argument which follows has been added to our Supplementary Information, and a line of text directing readers to it has been added to the discussion of the unpaired surface in our main text:

If we consider a simple layered system with alternating interlayer hoppings t_A and t_C , but which terminates with a vacuum beyond the top surface (see Figure R9 below), we can first imagine the case in which $t_A = t_C$ (left of Fig. R9), which clearly corresponds to a case of half-filling and a metallic state is realized. Note that this also corresponds to the purely T_A and purely T_C stacked 1T-TaS₂ cases, both of which are unsurprisingly predicted, using DFT calculations, to be metallic in the absence of electronic correlations [Ritschel *et al.*, PRB **98**, 195134 (2018)]. Now if the ratio $t_A/t_C > 1$ is varied, describing the case that the bulk of the structure becomes dimerized but the top layer is unpaired, we can proceed onwards to the extreme case in which $t_C \ll t_A$ (middle of Fig. R9), at which point it is clear that the top layer is completely decoupled, and again must be half-filled, and with an in-plane bandwidth determined by its own intra-layer hopping, which we can call $t_{||}$. As the ratio $t_A/t_C > 1$ is increased, we should expect the bulk of the material to pass through a metal-insulator transition. But importantly, *there is no point between these two extremes at which the metallic state in the uppermost layer will disappear.*

Figure R9. Ubiquity of surface metallicity in absence of electronic correlations, for a dimerized, layered system with an unpaired top layer.

The detailed behavior of this model is that the bulk will have a gap determined by the difference in inter-layer hoppings, $|t_A| - |t_C|$, and the surface layer will retain a metallic band of bandwidth $6t_{\parallel}$ (for a triangular lattice). Because the relevant orbitals are of d_{z^2} character, we can realistically expect t_A to be larger than either t_{\parallel} or t_C , and so the bandwidth $6t_{\parallel}$ may not necessarily span the dimerization gap $2\Delta = 2(|t_A| - |t_C|)$, but a half-filled band should nevertheless persist at the surface for any set of relative strengths of t_{\parallel} , t_A and t_C , as long as $t_A/t_C > 1$. (Note that even if $t_{\parallel} \rightarrow 0$, as shown in the figure, this model reduces to the SSH model mentioned by the Referee (right-hand-side of Fig. R9), which also supports an end state at zero energy for the unpaired site. In this sense, the importance of the additional extra dimensions in the above model is only that the end/surface state now has a bandwidth controlled by t_{\parallel} .)

Given that we should expect that $t_A/t_C > 1$ in the physical system under discussion (as thoroughly argued above), this provides a very strong argument that the best explanation for the observed gap is strong electronic correlations, and this fully justifies the use of the term ‘*Mottness*’ in our title.

In relation to the discussion here, we have added a note of Acknowledgment in our manuscript to thank Dr. Motoaki Hirayama and Dr. Ryotaro Arita for their invaluable advice.

The discussion of the origin of the spectral gap at the unpaired surface in the main text has been modified, and substantially streamlined, in order to mesh with the above argument, which

is reproduced in the Supplementary Information. (For the relevant revisions, please see Items 2, 3 and 7 in the List of Changes.)

2. Regarding the band bending effects, seeing the gap size increase with increased current seems quite normal. Others have seen this in other transition metal dichalcogenides. The idea here is that the ratio of the voltage drop between the first layer and second layer compared to the drop between tip and first layer increases as the tip gets closer. (This gets a bit complicated when the tip and sample have a large work function mismatch, where the contact potential dominates and VB and CB can, for example, bend in the same direction. I believe Feenstra's simulations used a large contact potential.) The discussion in the supplement should be modified to reflect this.

Here the Referee's comments are correct by themselves: Seeing the gap size increase with increased current would be quite normal, and we agree. However, in our data (Fig. R4 in our previous Reply Letter and reproduced below), the apparent gap size increases with *decreasing* current. The ratio of the voltage drop between the first and second layers to the drop between the tip and first layer should ordinarily increase as the tip gets closer, which, for positive bias, enhances the upward tip-induced shift for a given spectral feature. But what we observe is that the upward band-bending is apparently *suppressed* rather than enhanced as the tip gets closer, or even that there is a downward bending which is being enhanced. See below:

Figure R10. Tip-height-dependence of conductance spectra. Conductance peaks at the UHB onset for paired (a) and unpaired (b) clusters. In each case, the apex of the conductance

peak, observed at the energy E_0' clearly shifts towards higher energy as the tip-sample distance increases.

However, the Referee points out that the difference in tip and sample work functions plays an important role in the TIBB phenomenon, and this point is very helpful to us. We now realize that if the sample work function is significantly larger than that of the tip, the apparent tip-induced potential can be non-zero even for zero applied bias, and the condition of zero tip-induced bending can be found at a non-zero applied bias, corresponding to the energy $W_0 = W_{\text{sample}} - W_{\text{tip}}$. This is the case in a recent work by Battisti *et al.* [PRB **95**, 235141 (2017)], where it is shown that a negative tip-induced potential can be present for a small positive bias (see Fig. 3(e) of that paper). This could explain the observed downward bending which is enhanced by the proximity of the tip to the sample in our measurements. We have added a note of this in our Supplementary materials (please see Item 8 in the List of Changes).

We again sincerely thank the Referee for these very helpful comments, which have helped to solve a long-standing problem for us.

In light of the above however, we still re-iterate that regardless of the sign and underlying mechanism of this tip-induced shift, its effect is quite small (on the order of only a few meV in the regime typical for STM measurements, as seen in Fig. R10 above). It is not significant enough to impact on any of the conclusions drawn in this manuscript. We put forward our estimates of the band gap sizes for the respective surfaces as the best available so far, while making note of the above-mentioned tip-induced effects.

List of Changes

- 1) To improve readability, with respect to Fig 1e, an additional line (underlined below) has been added to the description of this data in the main text:

“Spectra showing a gap in the density of states (DOS) ~150 meV, broadly consistent with those shown in previous STM reports [9, 23-25] were observed on eighteen of the twenty-four surfaces (similar to the blue curve labelled ‘Type 1’ in Fig. 1e). The prominent conductance peaks at around 200 meV and -200 meV have usually been

identified with the upper and lower Hubbard bands (UHB and LHB), respectively, characteristic of the Mott insulating state. A different form of the DOS, with a smaller gap of 50~60 meV, was observed...

- 2) We have revised again the interpretation of the spectral gap at the unpaired surface, including a note to direct the reader to the comprehensive discussion which we have added to the Supplementary Information (see below). The revised text is as follows.

“The surface of unpaired SD clusters (Regions 1 & 4) represents a new and perhaps qualitatively distinct system, which may allow us to disentangle the role of strong e-e interactions from that of unit-cell doubling in the electronic structure. First, we note that if the BL-stacked bulk charge order can be considered as dimerised, then breaking the dimerization by terminating the structure with a layer of unpaired clusters should be expected to leave a metallic surface state, at least in the absence of e-e correlations. This remains true even if significant inter-layer hopping is present, as long as the intra-dimer hopping, in this case across the T_A -stacked interface, is greater than that across the T_C -stacked interface between the uppermost layer and those below (for a more comprehensive argument, see the Supplementary Information). Hence, the spectral gap observed in the unpaired SD layer, where an otherwise metallic surface state is expected, may be attributed to Mott localisation.”

- 3) A note of thanks to both Dr. Motoaki Hirayama and Dr. Ryotaro Arita have been added to our Acknowledgments section.
- 4) Additional funding information (KAKENHI grant no. JP19H05602) has been added in the Acknowledgments section.
- 5) In partial answer to the first Referee’s queries, and to satisfy any curious reader, Fig. R7 above, along with a suitable description, has been added to the Supplementary Information, showing the spatial distributions, indicated by conductance mapping, of the spectral features at each surface.

“In Fig. S2 below, we elucidate the spatial distributions of the various spectral features, which may help to understand how each one corresponds to the Mott-localised orbitals and the valence and conduction bands (VB and CB). The upper panel shows that the peaks on either side of the gap for the paired surface are localised at the cluster centres,

and the CB and VB appear as honeycomb-like patterns around the cluster peripheries. This is consistent with the spatially resolved conductance data shown previously by Qiao et al. [7].

As can be seen in the lower panel, the spectrum for the unpaired surface shows several differences from that of the paired surface, aside from the smaller energy gap. Most notable is the pair of peaks below the gap, at around -120 meV and -240 meV. The peak residing at the higher energy, -120 meV, might be tentatively recognised as the lower Hubbard band (LHB), but if so what is the origin of the lower lying of the two peaks? From the spatially resolved conductance at each energy, we see that both are localized at the cluster centres, so the lower lying of the two peaks should not be associated directly with the CDW formation or valence band. The reason is that orbitals which reconstruct upon formation of the SD superstructure are thought to be the twelve which lie around the periphery of each cluster, similar to the image at -450 meV. (In the work of Qiao et al., this much lower-lying feature is conservatively attributed to the valence band, not specifically to the CDW, and we follow in the same spirit.) From the localization of the two peaks at the cluster centers, it is reasonable that these peaks can be described as a 'split LHB' as the Referee suggests (and likewise for the UHB). However, from the observations presented here we are not able to establish the reason why the LHB or UHB may exhibit an energy splitting. We note that the spatial distributions of the CB and VB features are very similar for both surfaces, indicating that they probably reflect stacking-independent electronic structures."

- 6) Figure S3e (formerly S2e) in the Supplementary Information has been corrected in accordance with the first Referee's comment.
- 7) The comprehensive justification for the expectation of a metallic surface state in the absence of electronic correlations for the unpaired surface, as well as the accompanying Figure (Fig. R9 above) has been included in the Supplementary Information, and reads as follows:

"We consider a simple layered system with alternating interlayer hoppings t_A and t_C , but which terminates with a vacuum beyond the top surface, as shown in Fig. S8 below. We can first imagine the case in which $t_A = t_C$ (as on the left-hand side of Fig. S7), which clearly corresponds to a case of half-filling in which a metallic state is realized in the absence of electronic correlations. Note that this also corresponds to the purely T_A and

purely T_C stacked 1T-TaS₂ cases, both of which are unsurprisingly predicted, using DFT calculations without on-site repulsion U , to be metallic [4]. Now if the ratio $t_A / t_C > 1$ is varied, describing the case that the bulk of the structure becomes dimerised but the top layer is unpaired, we can proceed onwards to the extreme case in which $t_C \ll t_A$ (middle of Fig. S7), at which point it is clear that the top layer is completely decoupled, and again must be half-filled, and with an in-plane bandwidth determined by its own intra-layer hopping, which we can call t_{\parallel} . As the ratio $t_A = t_C > 1$ is increased, we should expect the bulk of the material to pass through a Peierls-type metal-insulator transition. But importantly, there is no point between these two extremes at which the metallic state in the uppermost layer will disappear.

The detailed behavior of this model is that the bulk will have a gap determined by the difference in inter-layer hoppings, $|t_A| - |t_C|$, and the surface layer will retain a metallic band of bandwidth $6t_{\parallel}$ (for a triangular lattice). Depending on the relative strengths of t_{\parallel} , t_A and t_C , the bandwidth $6t_{\parallel}$ may not necessarily span the dimerisation gap $2\Delta = 2(|t_A| - |t_C|)$, but a half-filled band should nevertheless persist at the surface for any set of relative strengths, as long as $t_A = t_C > 1$. (It is noteworthy that if $t_{\parallel} \rightarrow 0$, as shown on the right-hand-side of Fig. S7, this model reduces to the well-known Su-Schrieffer-Heeger (SSH) model, which also supports an end state at zero energy for the 'unpaired' site.)

Does the physical systems under discussion, namely the unpaired surface termination of 1T-TaS₂, correspond to a case in which $t_A = t_C > 1$? Ab initio calculations performed by Lee et al., indicate that the T_C -stacked interface, specifically, should cause sufficient suppression of out-of-plane bandwidth to open a gap overall for the bulk ACAC stacking structure [8]. This suggests that the Peierls-like dimerisation should be thought of as straddling the T_A -stacked interface, not the T_C -stacked interface, and therefore that $t_A > t_C$. This also matches well with the basic intuitions that (i) the larger absolute cluster-cluster distance across the T_C -stacked interface should lead to much smaller orbital overlap, and (ii) the lateral offset across the T_C -stacked interface should also lead to much smaller overlap due to the d_{z^2} character of the relevant orbitals [7]. Given that the system under discussion, the unpaired surface, should correspond to the case in which $t_A = t_C > 1$ in the above model, this provides a strong argument that a metallic surface state is expected if electronic correlations are absent, and that such correlations are therefore best explanation for the observed spectral gap.

- 8) The discussion of the tip induced band bending effect in the Supplemental information has been modified, as follows:

“The absence of strong tip-height-dependent TIBB energy shifts of the upper and lower band onsets indicates that these measurements reflect the intrinsic gap sizes reasonably well. Established methods such as that formulated by Feenstra et al. can in principle be used to account for the TIBB effect and ascertain a more accurate measurement of the gap size [8, 9]. In such a framework it is usual that in a semiconductor or insulator, a given unoccupied state ordinarily residing at E_0 should be observed at an energy E_0' which shifts increasingly higher as the tip-sample separation is decreased - i.e. $dE_0'/dz_{\text{offset}} < 0$. For this reason, it is typically thought that spectra acquired using STM overestimate the spectral gap size for a semiconductor or insulator. However, in this case there is reason for caution: Interestingly, in Figure S9 it can be seen that in fact $dE_0'/dz_{\text{offset}} > 0$ for the observed onsets of unoccupied states at both surfaces. Recently, Battisti et al have observed that such a tendency can arise in cases where the sample work function is significantly larger than that of the tip. The apparent tip-induced potential can be non-zero even for zero applied bias, and the condition of zero tip-induced bending can be found at a non-zero applied bias, corresponding to the energy $W_0 = W_{\text{sample}} - W_{\text{tip}}$, where $W_{\text{sample(tip)}}$ is the sample (tip) work function [10]. As in the case of Battisti et al, here it is likely that a negative tip-induced potential is present for a small positive bias. (Although the sample work function can be determined, generally the tip work function cannot.) Nevertheless, Fig. S9 shows that the TIBB effect results in an energy shift of only a few meV over the range of set-point parameters typical for STM measurements. For the combination of reasons described above, we do not anticipate a large TIBB effect which would impact any of the conclusions drawn in this work.”

- 9) After thorough proof-reading, several minor typos have been corrected, and minor clarifications added, throughout the Supplementary Information text.

Reviewers' comments:

Reviewer #1 (Remarks to the Author):

The authors have substantially improved the manuscript and addressed most of the issues raised. I appreciate the additional spatially-resolved spectra of split bilayer that elucidate the correlated origin of the spectral features. I still think that the qualitative splitting of the spectrum requires proper discussion prior to accepting the paper. Similarly, spectral feature assignment will improve the clarity of the message. I am happy to recommend the paper after adding the proper and self-consistent discussion in the main text.

Below I reiterate two questions from my previous review as well as the part of the authors reply and then provide my further comments.

Previous question 1a:

Please, assign the peaks in the red and blue curves to Hubbard bands and CDW bands respectively (as it was done in e.g. NatComm 7, 10956 (2016) or NatComm 7, 10453 (2016)).

Authors' reply to question 1a:

Please allow us to answer these two points together in the discussion below. In answer to the Referee's comment regarding the origins of particular spectral features, we are reluctant to definitively assign a particular peak to the CDW, even for the case of the paired cluster surface. Previous reports, which appear to measure the surface which we have demonstrated is the paired surface, have ventured to identify certain spectral peaks with the CDW, but these have appeared variously at around -375 meV [Ma et al., Nat. Comms 7, 10956 (2016)], around -450 meV [Cho et al., Nat. Comms 7, 10453 (2016)], and even as low as -650 meV [Zhu et al., PRL 123, 206405 (2019)]. Therefore, even the commonly observed paired surface is not as well understood as it may seem from any one paper, and more generally, STM observations alone are probably not well suited to definitively argue the underlying mechanism behind such features.

New comment to 1a:

I tend to disagree with the authors comment here. There are overwhelming observations that allow to assign the spectral features to Hubbard and CDW bands. One can mention the classic STM work [Kim et al. PRL 73, 2103 (1994)] that shows the transfer of spectral weight with temperature upon crossing the NC to C + Mott transition. Furthermore, ARPES shows that Hubbard and CDW bands are located in different parts of Brillouin zone [see e.g. Rossnagel J. Phys. Cond Mat. 23, 213001 (2011)]. Even more important, time-resolved ARPES [see e.g. Hellman et al., Nat. Comm. 3, 1069 (2012); Ligges et al., PRL 120, 166401 (2018)] reveals qualitatively different dynamics of these bands, linking the characteristic timescale of the CDW gap closing with the CDW amplitude mode frequency. Finally, Ma et al. [Nat. Comms 7, 10956 (2016)] made further attempts to compare quantitatively ARPES and STM results. The variation in sizes etc. in different papers is certainly the problem to be discussed, but cannot prevent the qualitative ascribing of the bands. With the above said, I cannot understand the authors' reluctance to ascribe the bands, as it reduces the uncertainty in understanding their results and supports their claim of correlated physics involved (CDW bands are intact as shown below). For the reasons described above and the references provided, I am adamant in my request.

Previous question 1c:

The spectrum of the unpaired layer seems to have two split Hubbard bands (e.g. the two peaks around -200meV in the red curve are symmetrically shifted with respect to the original LHB in the blue curve). Could it be that the Hubbard bands are split due to the two inequivalent hopping integrals that appear due to the asymmetry of the David star position in the unpaired layer with respect to those in the underlying paired layer?

Authors' reply to question 1c:

[...]

Please note that a TC-stacked SD cluster actually has three underlying 'nearest neighbors' (for a very loose definition of 'nearest neighbors'). The three sites below all have slightly different absolute distances. However, the TC-stacked site is very close to the point which is equidistant to the three neighbors below (just slightly more than half of one Ta-Ta lattice constant away). Therefore, all three of the interlayer hopping integrals across the TC interface are probably very similar, and we do not expect the difference between them to be the cause of the apparent additional peak in the conductance spectrum.

New comment to 1c:

I have to respectfully disagree with the last sentence. Simple construction shows (see figure attached) that the three nearest (in the geometric sense) neighbors are located at distances: $B_{1} = \sqrt{4a^{2} + c^{2}$, $B_{2} = \sqrt{(3a^{2} + c^{2})}$ and $B_{3} = \sqrt{7a^{2} + c^{2}$. The CDW period is $A_{CDW} = a\sqrt{13}$. This allows us to write down:

$A_{CDW} > B_{3} \gg B_{1} > B_{2};$

and taking into account numerical values for a and c:

$A_{CDW} \sim B_{3};$

$B_{1} \sim B_{2}$.

The distances enter the hopping integrals exponentially (further approximation can take into account Wannier orbital shape). From here it is already clear that there is a set of hopping integrals that will determine fine features. Qualitatively, there are two limiting hopping integrals: one is related to $A_{CDW} \sim B_{3}$ and another – to $B_{1} \sim B_{2}$. The role of this contribution to the splitting should be discussed in the manuscript.

Next, the description should be self-consistent. The authors argue that the change of stacking from TC to TB causes Mottness collapse. TB stacking gives: $A_{CDW} > B_{3} > B_{1} \gg B_{2}$, where $B_{2} = \sqrt{a^{2} + c^{2}}$ (see attached figure for construction). The B_2 change is minor on going from TB to TC. The only other thing that changes is the asymmetry in the David star position in the split bilayer with respect to the underlying full bilayer (expressed in the distances hierarchy). Which of these two factors is responsible for the Mottness collapse then? On the qualitative level, it is highly likely that the smaller hopping in the TC case being not enough to collapse Mott state still modifies it by splitting the band and reducing the gap.

Authors' reply to question 1 in general:

To conclude, within the scope of this manuscript, we contend that the important observations are, in brief, (i) that we have experimentally established that the alternating 'ACAC' stacking ('unit-cell doubling' in the title) is realized in 1T-TaS₂, and that (ii) despite this unit cell doubling, we still see signs of a correlation-driven gap at the surface where the doubling is broken. (A more thorough argument for this has now been added. Please see our reply to the second Referee.) The detailed mechanisms determining the differing spectral shapes at each surface are certainly interesting and deserve investigation, but are beyond the intended scope of this manuscript, and beyond the suitability of the tools used here to elucidate confidently.

New comment:

The qualitative features should be discussed and are within the scope of the paper for the reasons described in the two comments above.

TC stacking

Side view

Cut of the schematic in Figure 3a with the red and blue dashed ovals added on top to highlight the David star clusters in split bilayer on the normal bilayer respectively. \mathbf{B}_1 and \mathbf{B}_2 are the vectors connecting the David star centers where Hubbard bands are localized.

Top view

From the top view we can see that the David star cluster in the split bilayer (red) has three nearest neighbors in the normal bilayer (blue) underneath. The vectors between the centers (where Hubbard bands are localized) are:

$$\mathbf{B}_1 = 2\mathbf{a} + \mathbf{c}; |\mathbf{B}_1| = \sqrt{4a^2 + c^2}$$

$$\mathbf{B}_2 = \mathbf{a} + \mathbf{b} + \mathbf{c}; |\mathbf{B}_2| = \sqrt{3a^2 + c^2}$$

$$\mathbf{B}_3 = 2\mathbf{a} + \mathbf{b} + \mathbf{c}; |\mathbf{B}_3| = \sqrt{7a^2 + c^2}$$

CDW period $\mathbf{A} = 3\mathbf{a} + \mathbf{b} = a \cdot \sqrt{13}$; \mathbf{a} , \mathbf{b} – in-plane atomic unit vectors; \mathbf{c} – out of plane atomic unit vector.

One can clearly see that $B_1 \sim B_2$ and $B_3 \sim$ CDW period: $A > B_3 \gg B_1 > B_2$. Either way, two qualitatively distinct hopping integrals emerge.

TB stacking

Side view

Cut of the schematic in Figure 3e with the red and blue dashed ovals added on top to highlight the David star clusters in split bilayer on the normal bilayer respectively. B_2 is the vectors connecting the two David star centers (one on top of the other) where Hubbard bands are localized.

Top view

From the top view we can see that the David star cluster in the split bilayer (red) has three nearest neighbors in the normal bilayer (blue) underneath. The vectors between the centers (where Hubbard bands are localized) are:

$$\mathbf{B}_1 = 2\mathbf{a} + \mathbf{b} + \mathbf{c}; |\mathbf{B}_1| = \sqrt{7a^2 + c^2}$$

$$\mathbf{B}_2 = \mathbf{a} + \mathbf{c}; |\mathbf{B}_2| = \sqrt{a^2 + c^2}$$

$$\mathbf{B}_3 = 3\mathbf{a} + \mathbf{c}; |\mathbf{B}_3| = \sqrt{9a^2 + c^2}$$

CDW period $\mathbf{A} = 3\mathbf{a} + \mathbf{b} = a \cdot \sqrt{13}$; \mathbf{a} , \mathbf{b} – in-plane atomic unit vectors; \mathbf{c} – out of plane atomic unit vector.

One can clearly see that $B_2 \sim B_3 \sim \text{CDW period}$ and $A > B_3 > B_1 \gg B_2$.

Reviewer #2 (Remarks to the Author):

In the new version of their paper, Butler et al. have addressed my major concerns with additional data and explanations. I now recommend publication in Nature Communications. The manuscript is quite nice and reports a very interesting result.

Reply Letter (round 3)

We sincerely thank both Referees again for their continued careful reading and helpful comments, including Referee 2 for the endorsement of our manuscript for publication.

Below we respond to the two remaining comments from Referee 1, which have again helped us to improve the clarity of our message, and the manuscript as a whole. The Referee's comments are reproduced below in red, and as before, figure labels are continued on from our previous reply letter. This reply is again followed by a List of Changes, where revisions are listed in order of their appearance in the manuscript.

“New comment to 1a:

I tend to disagree with the authors comment here. There are overwhelming observations that allow to assign the spectral features to Hubbard and CDW bands. One can mention the classic STM work [Kim et al. PRL 73, 2103 (1994)] that shows the transfer of spectral weight with temperature upon crossing the NC to C + Mott transition. Furthermore, ARPES shows that Hubbard and CDW bands are located in different parts of Brillouin zone [see e.g. Rossnagel J. Phys. Cond Mat. 23, 213001 (2011)]. Even more important, time-resolved ARPES [see e.g. Hellman et al., Nat. Comm. 3, 1069 (2012); Ligges et al., PRL 120, 166401 (2018)] reveals qualitatively different dynamics of these bands, linking the characteristic timescale of the CDW gap closing with the CDW amplitude mode frequency. Finally, Ma et al. [Nat. Comms 7, 10956 (2016)] made further attempts to compare quantitatively ARPES and STM results. The variation in sizes etc. in different papers is certainly the problem to be discussed, but cannot prevent the qualitative ascribing of the bands.

With the above said, I cannot understand the authors' reluctance to ascribe the bands, as it reduces the uncertainty in understanding their results and supports their claim of correlated physics involved (CDW bands are intact as shown below). For the reasons described above and the references provided, I am adamant in my request.”

We are now convinced that it is appropriate to label the peaks in spectral weight observed at -300 meV and below as originating from the CDW reconstruction, especially after reading in more detail the work of Rossnagel et al. [J. Phys.: Condens. Matter **23**, 213001 (2011)]. This labelling has now been adopted in our Figure S2 in the Supplemental Information.

With regard to Fig. 1e, as stated in our earlier reply, it is not a suitable place to label the CDW features, because those spectra were collected at SD cluster centers, and therefore reflect predominantly the Mott-localised features. The images shown in Fig. S2 show that, in contrast, the local density of states associated with the CDW lies around the periphery of each cluster. For this reason, we can fully label the CDW and Hubbard peaks only in Fig. S2, where curves collected at the cluster centers and peripheries are shown together. The labels in Fig. S2 have been altered accordingly, and the work by Rosnagel *et al.* has been cited.

Please note that although we don't label the UHB and LHB within Fig. 1e, we have added the following text to the caption, for clarity (Item 1 in the List of Changes):

“The prominent conductance peaks at around 200 meV and -200 meV in the previously reported Type 1 spectrum have usually been identified with the upper and lower Hubbard bands.”

As well as the above, we add a citation to the seminal STM work by Kim *et al.* [PRL **73**, 2103 (1994)] to the suitable point in the main text attributing the peaks above and below the gap to the UHB & LHB (Item 2 in the List of Changes).

We also add this citation and an additional line to the discussion of our conductance maps of the CDW, UHB and LHB in the Supplementary Information (Item 6 in the List of Changes):

“The positive spatial correlation between the spectral features at 200 and -200 meV attests to their identification as Mott localised orbitals [6].”

“New comment to 1c:

I have to respectfully disagree with the last sentence. Simple construction shows (see figure attached) that the three nearest (in the geometric sense) neighbors are located at distances: $B_1 = \sqrt{(4a^2 + c^2)}$, $B_2 = \sqrt{(3a^2 + c^2)}$ and $B_3 = \sqrt{(7a^2 + c^2)}$. The CDW period is $A_{CDW} = a\sqrt{13}$. This allows us to write down: $A_{CDW} > B_3 \gg B_1 > B_2$; and taking into account numerical values for a and c : $A_{CDW} \sim B_3$; $B_1 \sim B_2$. The distances enter the hopping integrals exponentially (further approximation can take into account Wannier orbital shape). From here it is already clear that there is a set of hopping integrals that will determine fine features. Qualitatively, there are two limiting hopping integrals: one is related to $A_{CDW} \sim B_3$ and another - to $B_1 \sim B_2$. The role of this contribution to the splitting should be discussed in the manuscript. ...”

The Referee's suggestion is well taken. After considering the Referee's construction, and after further discussion with our theorist colleagues, we consider that a detailed calculation of the various energy scales associated with the hierarchy of overlaps is needed in order to draw even qualitative conclusions about the detailed spectral shape. An explicit calculation for such complex clusters is exceptionally challenging. However, we agree with the Referee's suggestion to discuss the possible origin of the spectral details. To that end, the following line has been added to the discussion (Item 3 in the List of Changes):

"It is possible that the anisotropic coordination environment of the T_C -stacked clusters results in a hierarchy of varying nearest-neighbour orbital overlaps which could explain the detailed spectroscopic features observed. The resulting modification of bandwidth may also be the cause of the apparent reduction of the Mott gap."

Nevertheless, as this remains unresolved, we retain the question posed in the final paragraph of the manuscript:

"Questions arise about the detailed mechanisms in play at the distinct surfaces observed, as well as between layers in the ACAC-stacked bulk - Why does the surface of unpaired clusters have a smaller gap than the paired layer, and what explains the details of its spectral shape?"

As an aside, we also include a line of text which adds to the interpretation for the nature of the spectral gap at the *paired* surface, citing an additional reference which we see as very important for discussions which may follow on from ours, in light of the experimental observation of bilayer-type stacking (the modified or newly added text is underlined):

"They also indicate that while the BL-stacked bulk structure of $1T-TaS_2$ may satisfy the criteria for a simple band insulator [5, 7], this does not preclude the presence of strong e-e correlations, and these have been evidenced in the recent observation of doublon excitations characteristic of a Mott state [6]. Indeed, it has previously been shown that a system described by a two-layered Hubbard model with inter-layer hopping t_{\parallel} can exhibit a continuous crossover between the Mott and band insulating regimes [32]. Hence, for the paired surface and the BL-stacked bulk, with significant intra-BL orbital overlap, the two regimes may not be meaningfully distinct."

The newly added reference is the following:

*[32] Fuhrmann, A., Heilmann, D. & Monien, H. From Mott insulator to band insulator: A dynamical mean-field theory study. Phys. Rev. B **73**, 245118 (2006).*

Please see items 4 in the List of Changes.

... Next, the description should be self-consistent. The authors argue that the change of stacking from TC to TB causes Mottness collapse. TB stacking gives: $A_{CDW} > B_3 > B_1 \gg B_2$, where $B_2 = \sqrt{a^2 + c^2}$ (see attached figure for construction). The B_2 change is minor on going from TB to TC. The only other thing that changes is the asymmetry in the David star position in the split bilayer with respect to the underlying full bilayer (expressed in the distances hierarchy). Which of these two factors is responsible for the Mottness collapse then? On the qualitative level, it is highly likely that the smaller hopping in the TC case being not enough to collapse Mott state still modifies it by splitting the band and reducing the gap.

The referee mentions two possible effects which could lead to Mottness-collapse upon going from \mathbf{T}_C to \mathbf{T}_B . The first is the reduction of the smallest nearest-neighbour (n - n) distance in the distance hierarchy, and the second is related to increased asymmetry (increasing imbalance between inter-cluster hoppings). The Referee seems to dismiss the first reason, suggesting that the change in the smallest n - n distance is minor. Below, by numerically evaluating a simple model we find that, on the contrary, the change in n - n distances could plausibly cause a bandwidth enhancement of the scale required to explain the Mottness-collapse.

As a first approximation, we may consider that the overlap integral for neighbouring orbitals is an exponentially decaying function of their absolute distance (in an s-wave approximation). The numerical result shown in Fig. R11, using $|\mathbf{a}| = |\mathbf{b}| = 0.34$ nm and $|\mathbf{c}| = 0.6$ nm, verifies that if all three entries in the n - n distance hierarchy are included in a sum of exponentially decaying overlaps, then for any reasonable value of the decay constant κ , the total overlap for \mathbf{T}_B stacking is always larger than that for \mathbf{T}_C . (A 'reasonable' value of κ means that decay length-scale $1/\kappa$ is always larger than that for \mathbf{T}_C . (A 'reasonable' value of κ means that decay length-scale $1/\kappa$ is realistically short - no more than 0.1 nm or so. In the limit of large $1/\kappa$ the total overlap tends towards the coordination number, in this case three, regardless of stacking vector.) Please note that we identify and label the n - n distances in a different way from that used by the Referee, and that here B_i (C_i) denote the n - n distances for \mathbf{T}_B (\mathbf{T}_C) stacking.

Figure R11. Estimation of inter-layer orbital overlaps for three n - n , in an s-wave approximation, for T_B and T_C stacking configurations. (a) and (b) show the in-plane projected n - n distance hierarchies in each case. (c) shows a comparison of the respective sums of the n - n overlaps for each case, given that each n - n distance enters into an exponential decay characterised by κ . The total overlap for T_B stacking always exceeds that for T_C stacking in the relevant range of κ , as shown by the ratio of the sums (blue curve).

This simple model yields greater overlap for T_B than for T_C , qualitatively consistent with the suggested mechanism of Mottness-collapse upon transition from T_C to T_B .

From the idea that overlap is larger for T_B than for T_C , it clearly follows that the overlap would be maximised for T_A stacking of an unpaired layer on the underlying bilayer. This leads to a tentative prediction: That such a T_A stacked, unpaired layer atop a bilayer, yielding a top-down stacking order of AACAC... would be metallic at the surface due to the breakdown of the layer dimerisation (-the number of electrons per surface unit-cell is odd-) and the large inter-layer overlaps. This may be related with the prediction of an out-of-plane (OOP) metal for purely T_A -stacked clusters [Darancet, *et al.*, PRB **90**, 045134 (2014), Ritschel *et al.*, PRB **98**, 195134 (2018)]. However, this stacking configuration remains unobserved so far. The proposed consequences of progressively increasing inter-layer overlap are depicted in Fig. R12 below.

Fig. R12. Consequences of increasing inter-layer overlaps for various surface stacking configurations. Clusters in stacking sites which do not belong to the natural bulk stacking order are shaded in gray.

Given the simplified framework described above, the argument for Mottness-collapse at the T_B -stacked surface fits self-consistently: Due to change in n - n distances, the inter-layer overlap, and therefore bandwidth, is expected to be larger for T_B stacking than for T_C stacking, potentially explaining the observed Mottness-collapse.

An alternative mechanism -- the second mechanism mentioned by the Referee -- is that if the asymmetry of the stacking coordination and resulting hierarchy of n - n overlaps could modify the bandwidths and reduce (or close) the Mott gap. If this is the case for the T_C -stacked layer, as mentioned above, then the increased asymmetry of the T_B stacking coordination (see Fig. R11) might come with a relatively larger bandwidth enhancement, closing the gap. As mentioned above, a detailed calculation of the various energy scales associated with the hierarchy of overlaps is needed in order to draw conclusions about the peaks' splitting or bandwidths. An explicit calculation for such complex clusters is very challenging, and the contribution of this alternative mechanism is much harder to estimate, even qualitatively.

Ultimately, we are considering the hopping between orbitals centered in complex 39-atom clusters (if the top and bottom S layers are included, which have been neglected in the discussion so far), each with intricate internal bonding and orbital textures. The simplistic arguments we present above, or even explicit Slater-Koster-based tight-binding models which reduce the system only to the idealised $5d_{z^2}$ orbitals at the cluster centers, will clearly not be satisfying. A sufficiently detailed modelling of this complex system cannot be presented here, and we leave that as a target for future projects.

These arguments are now referred to in the main text, and largely reproduced in the Supplemental Information (see Item 7 in the List of Changes).

“The qualitative features should be discussed and are within the scope of the paper for the reasons described in the two comments above.”

Each of the Referee's concerns have been discussed in the revised version of the manuscript. We hope the revised version removes any remaining obstacles to acceptance of the manuscript.

List of Changes

- 1) The usual identification of the two conductance peaks of the Type 1 spectrum have been added to the caption of Figure 1.

“The prominent conductance peaks at around 200 meV and -200 meV in the previously reported Type 1 spectrum have usually been identified with the upper and lower Hubbard bands.”

- 2) Citation added to the main text:

*Kim, J.-J., Yamaguchi, W., Hasegawa, T. and Kitazawa, K. Observation of Mott Localization Gap Using Low Temperature Scanning Tunneling Spectroscopy in Commensurate 1T-TaS₂. Phys. Rev. Lett. **73**, 2103 (1994).*

- 3) A brief discussion of the detailed spectral shape of the Type 2 spectrum has been added in the discussion section of the main text:

“It is possible that the anisotropic coordination environment of the T_C-stacked clusters results in a hierarchy of varying nearest-neighbour orbital overlaps which could explain the detailed spectroscopic features observed. The resulting modification of bandwidth may also be the cause of the apparent reduction of the Mott gap.”

- 4) Additional discussion of the nature of the spectral gap at the paired (Type 1) surface has been included in the discussion section of the main text.

“They also indicate that while the BL-stacked bulk structure of 1T-TaS₂ may satisfy the criteria for a simple band insulator [5, 7], this does not preclude the presence of strong e-e correlations, and these have been evidenced in the recent observation of doublon excitations characteristic of a Mott state [6]. Indeed, it has previously been shown that a system described by a two-layered Hubbard model with inter-layer hopping t_{\parallel} can exhibit a continuous crossover between the Mott and band insulating regimes [32]. Hence, for the paired surface and the BL-stacked bulk, with significant intra-BL orbital overlap, the two regimes may not be meaningfully distinct.”

In relation to the above, an additional reference is included:

*[33] Fuhrmann, A., Heilmann, D. & Monien, H. From Mott insulator to band insulator: A dynamical mean-field theory study. Phys. Rev. B **73**, 245118 (2006).*

5) Reference 16 (Murayama *et al.*) and Reference 22 (Stahl *et al.*) have been updated to refer to the journal versions, in *Physical Review Research* and *Nature Communications* respectively, rather than the arXiv versions.

6) A line has been added to the discussion of the spatially resolved conductance maps in Fig. S2:

“The positive spatial correlation between the spectral features at around 200 meV and -200 meV attests to their identification as Mott localised orbitals [6].”

7) To establish a self-consistent explanation of Mottness-collapse at the T_B -stacked surface, the following discussion and figure have been added to the Supplemental Information.

“Here we discuss a possible mechanism for the apparent collapse of the Mott state, leading to a metallic electronic structure, upon transition of the uppermost unpaired layer from T_C to T_B stacking.

We pay attention to the stacking-dependent inter-layer orbital overlaps. As a first approximation, we may consider that the overlap integral for neighbouring orbitals is an exponentially decaying function of their absolute distance (in an s-wave approximation). Let B_i (C_i) denote the n-n distances for T_B (T_C) stacking. A simple numerical calculation shown in Fig. S9, using $|\mathbf{a}| = |\mathbf{b}| = 0.34$ nm and $|\mathbf{c}| = 0.6$ nm, verifies that if all three entries in the nearest-neighbour (n-n) distance hierarchy are included in a sum of exponentially decaying overlaps, then for any reasonable value of the decay constant κ , the total overlap for T_B stacking is always larger than that for T_C . (A ‘reasonable’ value of κ means that decay length-scale $1/\kappa$ is realistically short - no more than 0.1 nm or so. In the limit of large $1/\kappa$ the total overlap tends towards the coordination number, in this case three, regardless of stacking vector.)”

Figure S9. Estimation of inter-layer orbital overlaps for three n-n, in an s-wave approximation, for T_B and T_C stacking configurations. (a) and (b) show the in-plane projected n-n distance hierarchies in each case. (c) shows a comparison of the respective sums of the n-n overlaps for each case, given that each n-n distance enters into an exponential decay characterised by κ . The total overlap for T_B stacking always exceeds that for T_C stacking in the relevant range of κ , as shown by the ratio of the sums (blue curve).

This simple model yields greater overlap for T_B than for T_C , qualitatively consistent with the suggested mechanism of Mottness-collapse upon transition from T_C to T_B .

From the idea that overlap is larger for T_B than for T_C , it clearly follows that the overlap would be maximised for T_A stacking of an unpaired layer on the underlying bilayer. This leads to a tentative prediction: That such a T_A stacked, unpaired layer atop a bilayer, yielding a top-down stacking order of AACAC... would be metallic at the surface due to the breakdown of the layer dimerisation (-the number of electrons per surface unit-cell is odd-) and the large inter-layer overlaps. This may be related with the prediction of an out-of-plane (OOP) metal predicted for purely T_A -stacked clusters [Darancet, et al., PRB **90**, 045134 (2014), Ritschel et al., PRB **98**, 195134 (2018)]. However, this stacking configuration remains unobserved so far. The proposed consequences of progressively increasing inter-layer overlap are depicted in Fig. R12 below.

Fig. S10. Consequences of increasing inter-layer overlaps for various surface stacking configurations. Clusters in stacking sites which do not belong to the natural bulk stacking order are shaded in gray.

Given the simplified framework described above, the argument for Mottness-collapse at the T_B -stacked surface fits self-consistently: Inter-layer overlap, and therefore bandwidth, is expected to be larger for T_B stacking than for T_C stacking, potentially explaining the observed Mottness-collapse.

However, a satisfactory understanding of the mechanism of Mottness-collapse must consider the hopping between orbitals centered in complex 39-atom clusters (if the top and bottom S layers are included, which have been neglected in the discussion so far), each with intricate internal bonding and orbital textures. The simplistic arguments we present above, or even explicit Slater-Koster-based tight-binding models which reduce the system only to the idealised $5d_{z^2}$ orbitals at the cluster centers, will clearly not be satisfying. A sufficiently detailed modelling of this complex system cannot be presented here, and we leave that as a target for future projects.”

- 8) A number of typos and minor grammatical errors in the Supplementary Information have been corrected.
- 9) In the Supplemental Information, φ was multiply defines as both the phase of the CDW order parameter and the angle in polar coordinates for CDW lattice displacement maps. This has been fixed. Now θ is used for the polar coordinates.

REVIEWERS' COMMENTS:

Reviewer #1 (Remarks to the Author):

The authors have satisfactorily replied to my comments. I appreciate the use of simple model and, at this point, I agree that more serious calculation, beyond the scope of this paper, is necessary to check the intuitive approach to the spectra.

I am happy to recommend this nice paper for publication.

Purely technical comment: I suggest using semi-log scale in Fig. S9c for clarity.

Reply Letter (round 4)

We again sincerely thank the Referee for the endorsement of the manuscript, and for the suggestion to revise Supplementary Figure 10. Below is a final List of Changes, describing the revision suggested by the Referee, and a point-by-point response to the Editorial requests. Newly added text is highlighted in blue.

List of Changes

Response to Referee's comments:

- 1) As suggested by the Referee, a semi-log scale has been used in Supplementary Figure 10c, which significantly improves the clarity:

Main text:

- 2) To shorten the Abstract below the 150-word limit, a few small changes have been made, while preserving the meaning. Please see the highlights and notes in the accompanying PDF. The Abstract length is now 148 words.
- 3) Section titles for the Introduction, Results and Discussion have been inserted.
- 4) At the end of the Introduction section, the following summary paragraph has been added: *"Here we report on low-temperature scanning tunnelling microscopy (STM) measurements which appear to confirm the premise described above: A unit-cell doubling inter-layer stacking pattern is indeed realised in 1T-TaS₂. Despite this, we see that a spectral gap persists at a surface where dimer-like inter-layer pairing is broken,*

which is unexpected unless e-e interactions play a significant role. We also show that for such an unpaired layer of SD clusters, a small change in stacking with respect to the underlying layer yields a metallic surface, suggesting that inter-layer effects underpin the microscopic mechanism of the material's metal-insulator transitions [23-29].”

- 5) With the addition described above, the abbreviation for STM is now introduced earlier in the text than in the previous version, and the following text has been altered accordingly.
- 6) The references [23-29] above, referring to metal-insulator transitions in 1T-TaS₂, had to be moved up the list. These are highlighted in the accompanying PDF.
- 7) Titles for subsections within the Results section have been inserted: ‘Observation of spectroscopically distinct surfaces’, and ‘Determination of inter-layer stacking’.
- 8) A small number of typos were fixed.

Language and style:

- 9) Speech marks have been removed around ‘Star-of-David’ and ‘dimerization’ in the Introduction, and also around ‘Mottness-collapsed’ in the Discussion.
- 10) Speech marks have been removed for ‘Type 1’, ‘Type 2’ and ‘ACAC’ upon introduction of each.
- 11) Italics on the contraction ‘e-e’ have been removed throughout.
- 12) In the Introduction, second paragraph, italics have been removed from ‘the electronic crystal’. Also, in the same sentence, speech marks have been removed from ‘cluster Mott insulator’.
- 13) For all instances of the stacking vectors $\mathbf{T}_{A,B,C}$ throughout the manuscript, italics for the subscripted labels have been removed. Italics for ‘ACAC’ have also been removed.

Italics have been removed for other subscripted labels, for example 'set' in I_{set} , in the Figure captions, and 'mod' in V_{mod} in the Methods section.

- 14) Near the beginning of the Discussion section, 'i.e.' is no longer in italics.

Methods and data:

- 15) Subsections have been added in the Methods section: 'Synthesis and low-temperature cleavage of samples', and 'STM measurements'.

End notes:

- 16) A 'Competing interests' statements has been added.
- 17) For Ref. 35, an arXiv preprint, the reference has been changed to: "Aishwarya, A. *et al.* Visualizing 1D zigzag Wigner crystallization at domain walls in the Mott insulator TaS₂. Preprint at <https://arxiv.org/abs/1906.11983> (2019)."
- 18) The complete page numbers (start--end) have been given for Ref. 37.

Display items:

- 19) The maths formatting described above has been applied to the Figure contents. For example, italics were removed on the subscripted label in Fig. 1c and Fig. 3e.
- 20) In the Figure captions, brackets have been removed from around the panel labels **a**, **b**, **c**...

Supplementary information:

- 21) Because each Supplementary Note should be referred to in the main text, an additional sentence has been added at the end of the section: 'Observation of spectroscopically distinct surfaces': 'The tip-height dependence of each of the spectra was investigated,

showing that there is no height dependent crossover between one type of spectrum and the other (see Supplementary Note 4).'

The Supplementary Notes and their Figures have now been re-ordered to match the order of their mentions in the main text. (Suppl. Note 11 becomes Suppl. Note 4, and the previous Suppl. Note 4 and onwards have been pushed down.)

- 22) For all mentions of 'the Supplementary Information', this has been replaced with a specific mention of the corresponding Supplementary Note or Supplementary Figure. Please see the highlighted PDF for details.
- 23) All of the above modifications regarding italics, math, labels, speech marks, etc. have been consistently applied throughout the Supplementary Information, and the Supplementary Figures and captions therein.
- 24) In the supplementary text and Supplementary Figure captions, a number of typos were fixed, and a few small modifications were made to the text, in order to improve rigor and clarity. No significant changes to the content were made.